# T cell receptor β-chains display abnormal shortening and repertoire sharing in type 1 diabetes

Iria Gomez-Tourino[1,2,3], Yogesh Kamra[1], Roman Baptista[1,2], Anna Lorenc[1] & Mark Peakman[1,2]

Defects in T cell receptor (TCR) repertoire are proposed to predispose to autoimmunity. Here we show, by analyzing $>2 \times 10^8$ *TCRB* sequences of circulating naive, central memory, regulatory and stem cell-like memory CD4[+] T cell subsets from patients with type 1 diabetes and healthy donors, that patients have shorter *TCRB* complementarity-determining region 3s (CDR3), in all cell subsets, introduced by increased deletions/reduced insertions during VDJ rearrangement. High frequency of short CDR3s is also observed in unproductive *TCRB* sequences, which are not subjected to thymic culling, suggesting that the shorter CDR3s arise independently of positive/negative selection. Moreover, *TCRB* CDR3 clonotypes expressed by autoantigen-specific CD4[+] T cells are shorter compared with anti-viral T cells, and with those from healthy donors. Thus, early events in thymic T cell development and repertoire generation are abnormal in type 1 diabetes, which suggest that short CDR3s increase the potential for self-recognition, conferring heightened risk of autoimmune disease.

[1] Department of Immunobiology, Faculty of Life Sciences & Medicine, King's College London, 2nd Floor, Borough Wing, Guy's Hospital, London SE1 9RT, UK. [2] National Institute for Health Research, Biomedical Research Centre at Guy's and St Thomas' Hospital Foundation Trust and King's College London, Guy's Hospital, London SE1 9RT, UK. [3] Present address: Immunology Laboratory, Biomedical Research Center (CINBIO), Centro Singular de Investigación de Galicia, University of Vigo, Campus Universitario de Vigo, Pontevedra 36310, Spain. Anna Lorenc and Mark Peakman jointly supervised this work. Correspondence and requests for materials should be addressed to M.P. (email: mark.peakman@kcl.ac.uk)

Type 1 diabetes (T1D) results from immune-mediated destruction of insulin-producing β-cells[1]. The vast majority of cases arise on a complex polygenic background, characterized by major disease-predisposing genes in the HLA region as well as much lower-risk allelic polymorphisms at >50 other immune gene loci (reviewed in ref. [2]). As a consequence, familial predisposition is a feature of T1D, especially when affected family members share HLA haplotypes[3], or are mono-zygotic twins[4,5]. However, reported disease concordance in such siblings and twins approximates only 50%[4,5]; thus beyond the currently known genes, there is a considerable gap in our understanding of what confers susceptibility to T1D. Whilst the interaction of environment and genes is a potentially key modifier of risk[5,6], there are as yet no concrete examples of this phenomenon, and, therefore, alternative propositions to account for missing heritability in T1D may be required.

One genetic element that cannot be revealed to be disease-linked in genome studies, but could nonetheless have considerable bearing on T1D risk, is the gene loci encoding the antigen-receptors borne by T and B lymphocytes. These receptors may confer the property of autoantigen recognition, fundamental bedrock of organ-specific autoimmune disease. For both cell types, the antigen receptor is generated by random somatic recombination of variable (V), diversity (D) and joining (J) genes, selected from germline encoded gene families[7]. Further diversity is engendered by variable degrees of nucleotide addition and deletion at the recombination sites, and by TCRα and TCRβ pairing[7]. As a result, highly diverse repertoires of antigen-receptors are generated at the level of the individual[8]. It is a guiding principle of autoimmune diseases such as T1D that antigen-receptors with the capacity to recognize self-antigens must be present within this repertoire in order for self-tolerance to be breached and autoimmune disease to develop. This is particularly true for the CD4+ T cell, which is viewed as the main initiator and driver of autoimmunity in T1D[9–11]. According to this model, CD4+ T cell recognition of β-cell autoantigens causes an unregulated autoimmune response, inflammation and destruction of islets of Langerhans, leading to clinical disease[9–11].

An attractive hypothesis to account for missing heritability in autoimmune disease is that risk is conferred by generation of an autoreactive-prone TCR repertoire. It has been proposed that TCR repertoires that are abnormally diverse (increases the potential for self-reactive TCRs), abnormally shared amongst individuals with disease (more likely that the repertoire contains CD4+ T cell clones with self-reactive, pathogenic potential), abnormally short (greater propensity to bind self HLA/peptide[12–17]) or abnormally constituted (higher hydrophobicity in the TCR-peptide-HLA interface seems to promote self-reactivity in animal models[18]) represent dangerous features that confer a risk of autoimmune disease that is additive to that conferred in the germline by HLA and other regions.

High-throughput sequencing technology and multi-dimensional fluorescence-based sorting of highly pure T cell subsets allow testing the hypothesis that features of the TCR repertoire within cellular compartments confers risk of auto-immune diseases. Here we present a large study of TCR repertoire in autoimmunity (>200 million sequences and 18 million unique clonotypes) performed on ex vivo sorted CD4+ T cell subsets. Patients with T1D have abnormally high TCR repertoire relatedness, shortness and interfacial hydrophobicity. Moreover, diversity, sharing and TCR clonotype length pre-thymic and post-thymic selection are correlated, and TCRB rearrangement patterns are aberrant in patients with T1D, suggesting a common underlying pathway for autoimmunity. Our results also demonstrate that TCRBs specific for autoantigens are shorter than their anti-viral counterparts, providing a link between self-reactivity and TCRB CDR3 length. These findings thus implicate abnormal thymic recombination as a predisposing factor for T1D.

## Results

### TCRB sequences show greater diversity in T cells from T1D.
The T cell receptor beta chain (TCRB) repertoires of different CD4+ T cell subsets (true naive, TN; central memory, CM; regulatory, Treg; and stem cell-like memory, Tscm) were examined using next generation sequencing technology in 14 recently diagnosed patients with type 1 diabetes (T1D) and 14 matched healthy donors (HD) who did not differ in mean age, distribution of gender, mean total cell number, cell subset yield or possession of HLA-DRB1*0401 or *0301 haplotypes associated with T1D (Supplementary Table 1; Supplementary Figs. 1a–e and 2a). The flow cytometric phenotype of sorted cell subsets was comparable between patients and healthy donors (Supplementary Fig. 2b–e). The number of cells per sorted subset correlated strongly with RNA yield (Spearman's $R = 0.75$; $p < 0.0001$) and a total of 107,917,701 cells were examined, yielding $>200 \times 10^6$ total and $>18 \times 10^6$ unique expressed TCRB sequences (productive unique sequences). There were no differences in the number of unique clonotypes obtained from patients and healthy donors for any of the four cell subsets (Supplementary Fig. 2f). Thus, in these experiments we sorted similar numbers of four major CD4+ T cell subsets from matched patient and control cohorts; cells had similar naive/memory flow cytometric phenotypes and yielded comparable numbers of unique clonotypes, allowing unbiased comparison of their TCRB repertoires.

As expected, higher numbers of sorted cells yielded more unique clonotypes (Fig. 1a). However, in the case of CM cells this relationship is asymptotic, indicating that in this subset we are close to sampling with sufficient depth to assess total diversity. It is also noteworthy that at equivalent numbers of sampled cells the CM subset is less diverse than TN (i.e., has fewer unique clonotypes), as might be predicted from the fact that CM cells undergo antigen-driven selection from the TN pool. To examine disease-related repertoire differences, normalized true diversity index and Gini coefficient (an index of clonality) were calculated for each of the samples (Fig. 1b, c), showing a trend for TN and CM cells from patients to be more diverse and less clonal, with reduced clonality being observed in TN cells in patients. Both diversity and clonality of Tregs are similar in the study groups, contrary to reports of reduced diversity in this subset in the non-obese diabetic mouse model[19]. Tscm cell diversity/clonality was similar between the groups. Interestingly, individuals with high diversity in the TN pool also have high diversity in the CM and Tscm pools (Fig. 1d–f), consistent with CM and Tscm propagating from TN. However, this does not apply to Treg cells (Supplementary Fig. 3), for which none of the diversity indices are correlated with those of corresponding Tconv cells.

### TCRB CDR3s are short and highly shared across T1D patients.
Abnormalities in diversity and clonality could result from deviations in complementarity-determining region 3 (CDR3) length and/or V and J gene usage (see below). Importantly, we identified a moderate, statistically significant reduction in TCRB CDR3 sequence length in patients with T1D in all four cell subsets (Fig. 2a–d. $p < 0.0005$ for TN and CM cells. $p < 0.001$ for Treg and Tscm cells. One-sided bootstrap univariate Kolmogorov–Smirnov test). To test whether these differences are not an artifact of sub-sampling of cells and variable number of reads among samples, we performed randomized sub-sampling analysis and re-analyzed the TCRB CDR3 length distributions. We find that, even in the subsamples, T1D patients maintain a

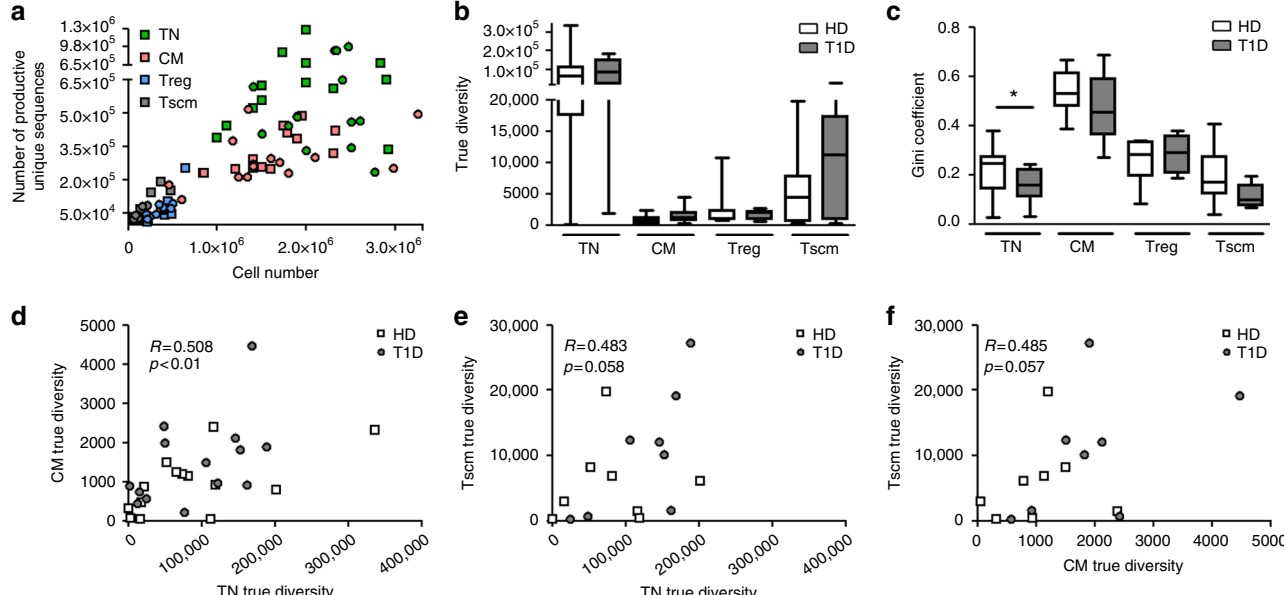

**Fig. 1** Diversity and clonality indices of TCRB repertoire of CD4[+] T cells. **a** The number of productive unique sequences (unique clonotypes) correlates with the cell number (Spearman's $R = 0.85$. $p < 0.0001$). **b**, **c** Sequencing data were normalized and true diversity indices **b** and Gini coefficients **c** were calculated (Methods section). TN cells from patients are less clonal than TN cells from healthy donors (Mann–Whitney U-test). Species is defined as a unique nucleotide sequence. **d**–**f** True diversity indices positively correlate among the TN, CM and Tscm cell subsets, consistent with CM and Tscm propagating from TN (Pearson's correlation). TN, true naive cells; CM, central memory cells; Treg, regulatory T cells; Tscm, stem cell-like memory T cells

statistically significant reduction in TCRB CDR3 length (Supplementary Fig. 4. $p < 0.05$. Mann–Whitney U-test). We reasoned that this biased TCRB CDR3 length could cause a higher degree of sequence sharing, as with shorter sequences (and hence more limited variability of sequence combinations), the likelihood of two TCRB CDR3s being identical just by chance increases. Indeed, it has been reported that highly shared TCR pools are enriched in clonotypes bearing lower numbers of insertions[20]. To examine this we calculated overlap indices for TCRB repertoires of all cell subsets. As shown in Fig. 2e–i and Supplementary Figs 5 and 6, TN, CM and Treg cell subsets are more similar among patients than amongst healthy donors. TN and CM cell repertoires are also more similar to each other in patients, consistent with propagation of CM from TN cells, in line with the observed correlation of the true diversity indices between these two cell subsets (Fig. 1d). Interestingly, we find a high degree of sharing of *TCRB* nucleotide sequences between CM and Treg cells in both healthy donors and T1D patients, supporting the concept that Treg cells may also be peripherally derived[21,22].

We next examined whether human leukocyte antigen (HLA) haplotypes influence *TCRB* sequence overlap. Interestingly, heterozygosity for the disease-predisposing haplotypes *DRB1\*03/DRB1\*0401/DQB1\*02/DQB1\*03* does not seem to result in a more similar thymic TCRB repertoire, as the TN/TN overlap indices are higher among patients with any other haplotype (Supplementary Fig. 7). This is not the case, however, for overlap among CM/CM, TN/CM and TN/Tscm cells (Supplementary Figs 7 and 8), where values are higher for *DRB1\*03/DRB1\*0401/ DQB1\*02/DQB1\*03* patients, pointing to these haplotypes having greater influence over the shaping of the peripheral TCRB repertoire.

The finding that diversity, sharing and length of the TCR repertoire are abnormal in T1D led us to examine the relationship between these metrics. Interestingly, we found that samples with high TN sharing also show lower clonality indices (Supplementary Fig. 9a), and higher frequencies of shorter TCR clonotypes (Supplementary Fig. 9b).

In summary, we find that length of TCRB clonotypes, their diversity and sharing are related, and each of these is abnormal in T1D, suggesting a common altered underlying pathway.

**Pre-selection T1D TCRB repertoires are also abnormal**. T cells with short TCRs could be a feature of the pre-selection TCR rearrangement process; or could undergo preferential enrichment during thymic negative and positive selection on self HLA; or a combination of the two. To gain a better understanding of which of these pathways is involved, we analyzed out-of-frame (OOF) sequences of TN cells as a means of studying the pre-selection TCR repertoire, as described elsewhere[20,23,24]. OOF sequences do not translate into functional TCRB chains, and cannot, therefore, directly influence TCR selection (Supplementary Fig. 10). Under these circumstances, the frequencies of insertions, deletions and *VDJ* gene usage can be examined without the confounding effects of positive and negative selection. By further restricting this analysis to TN cells, this approach has the additional advantage of ensuring that the results are also not biased by antigen-specific expansions.

Remarkably, and recapitulating our findings in productive *TCRB*s in TN, CM, Treg and Tscm pools, pre-selection TCRB CDR3s in patients with T1D are also shorter than those from healthy donors (Fig. 3a) and display a greater degree of sequence similarity (Fig. 3b–d). Interestingly, and as seen previously for productive sequences (Supplementary Fig. 9), individuals with higher percentages of short pre-selection clonotypes also show high sharing indices (Fig. 3e) and lower clonality indices (Fig. 3f) in the pre-selection repertoire.

Therefore, we examined whether the previously described alterations in length, diversity and sharing in the TN post-selection TCR repertoire in T1D patients are related to pre-selection alterations. Indeed, we find a strong positive correlation between pre and post-selection TCRB CDR3 lengths (Fig. 4a), with shorter TCRB CDR3s being enriched during thymic selection (Fig. 4b), a finding consistent with previous

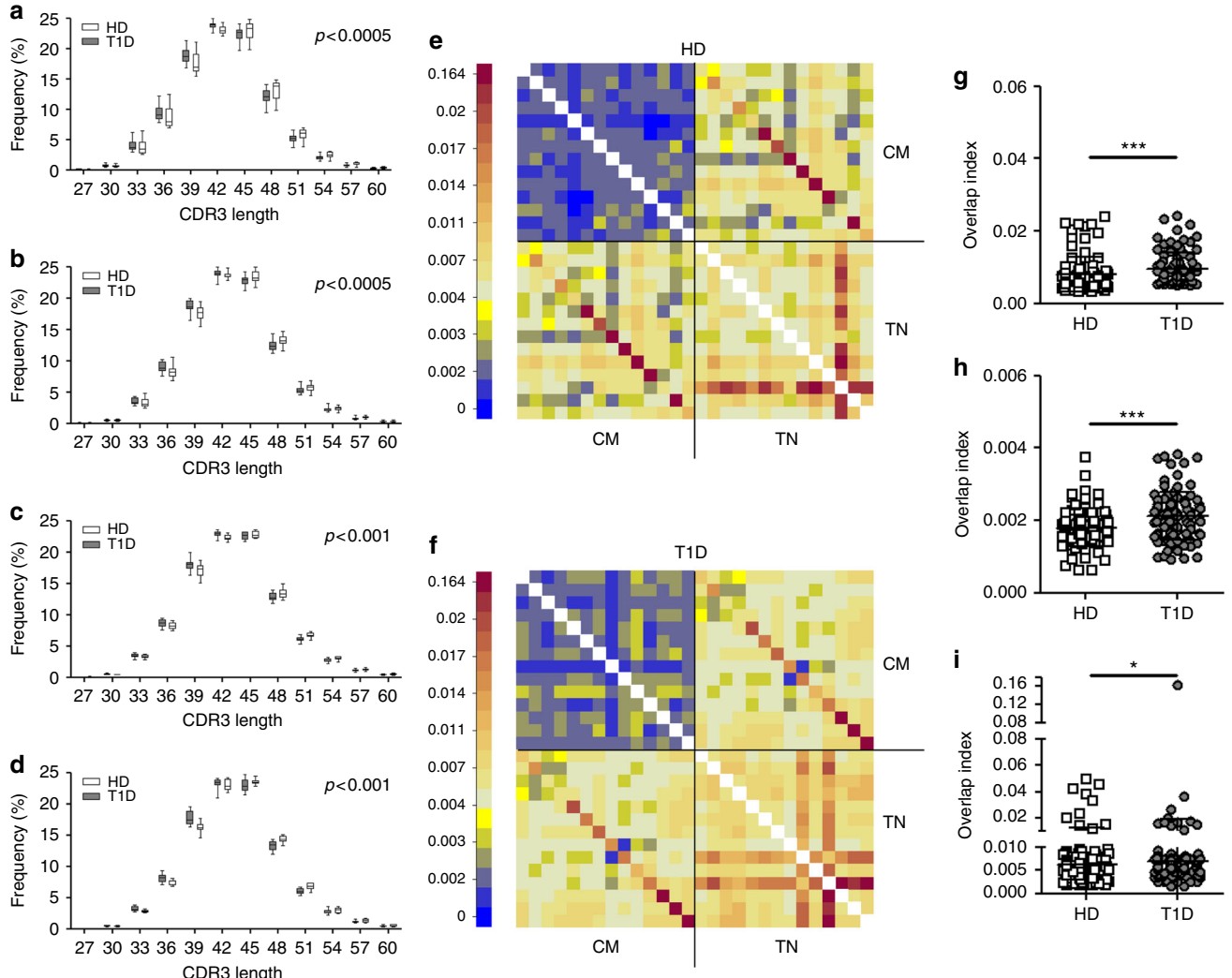

**Fig. 2** TCRB CDR3s are shorter and show greater similarity among T1D patients. **a–d** TCRB CDR3 nucleotide length of productive unique sequences from true naive (TN) **a**, central memory (CM) **b**, regulatory (Treg) **c** and stem cell-like memory (Tscm) **d** T cells present biased length distribution in patients, represented by higher frequencies of short TCRB CDR3s (and lower frequencies of long ones) (one-sided bootstrap univariate Kolmogorov–Smirnov test). **e**, **f** Overlap indices were calculated for TN and CM TCRB nucleotide repertoires from 14 healthy donors **e** and 14 T1D patients **f** as described in the Methods section, showing higher values among patients. **g–i** Overlap indices for TN/TN **g**, CM/CM **h** and TN/CM **i** cell subtypes (Mann–Whitney U-test). *$p < 0.05$; ***$p < 0.001$. Lines represent the mean $\pm$ SD

reports[12–17,25]. It is important to note, however, that patients and controls have similar degrees of enrichment of shorter clonotypes (Fig. 4c). Therefore, the bias towards shorter TCRB CDR3s in T1D patients must result from a higher input of shorter clonotypes into thymic selection.

These data imply that abnormalities in the pre-selection repertoire could underlie similar findings in the post-selection repertoire. Indeed, this concept is borne out by a series of additional observations: (i) samples with higher proportions of shorter pre-selection TCRB CDR3s show higher sharing and lower clonality in the post-selection repertoire (Fig. 4d, e); (ii) the degree of pre-selection sharing positively correlates with the amount of sharing and length distribution after selection ($R = 0.70$, $p = 3.59 \times 10^{-5}$; and $R = 0.48$, $p = 0.009$, respectively. Pearson correlation); (iii) true diversity and Gini coefficient pre-selection indices strongly correlate with their corresponding post-selection values (Fig. 4f, g); (iv) individuals with lower pre-selection clonality indices also have higher TCR repertoire sharing after selection (Fig. 4h).

In summary, we have shown that T1D patients present alterations in the pre-selection TCRB repertoire, including an increased frequency of shorter TCRB CDR3s, which undergo enrichment during thymic selection. These pre-selection alterations are perpetuated into the post-selection TN TCRB repertoire.

**Pre-selection TCRB rearrangement events are abnormal in T1D.** Our data thus far show that TCRB CDR3 length abnormalities are predominantly influenced by pre-selection events. Short TCRB CDR3 lengths might arise from reduced insertions and/or increased deletions during the TCRB rearrangement process (see Supplementary Fig. 11). Therefore, to understand the molecular basis for these features, we analyzed recombination events (indels) in each of the 6 rearrangement positions (Vdels, Jdels, D5dels, D3dels, n1ins and n2ins) (Fig. 5, and Supplementary Fig. 12). The distributions of frequencies of indels in pre- versus post-selection sequences indicate that TCRBs bearing excessively long Vdels or long n1ins are culled (Fig. 5a, e; compare left and right panels). In contrast, events in the *D–J* junction do not seem

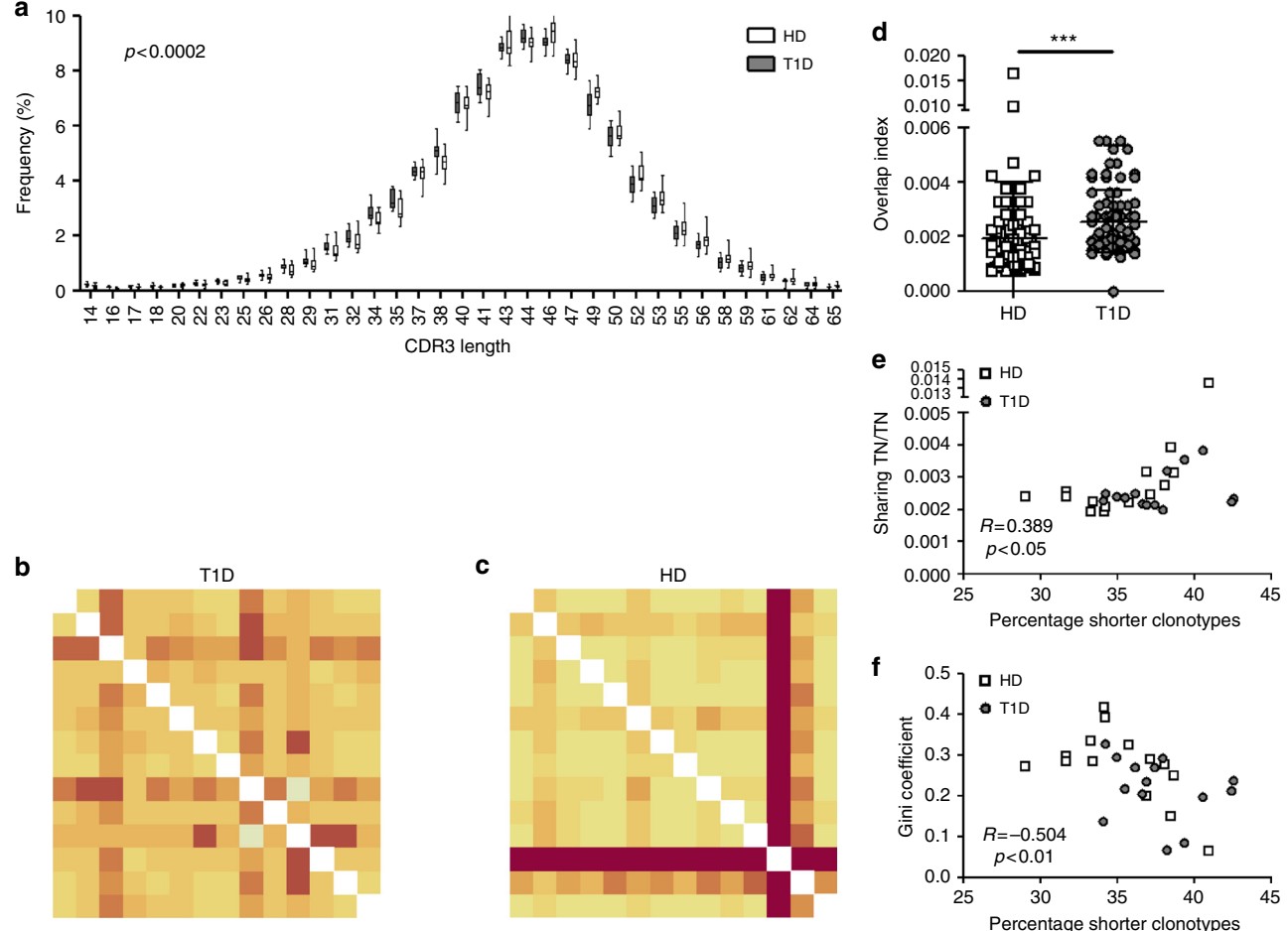

**Fig. 3** Pre-selection *TCRB* sequences are shorter and more similar among type 1 diabetes patients. **a** Distributions of TCRB CDR3 length (in nucleotides) of pre-selection unique sequences show a bias towards shorter lengths in type 1 diabetes (T1D) patients (one-sided bootstrap univariate Kolmogorov–Smirnov test; $p < 0.0002$). **b**, **c** Overlap indices for pre-selection unique nucleotide *TCRB* sequences are higher in T1D patients (**b**) than in healthy donors (**c**). **d** Overlap indices shown in **b**, **c** (Mann–Whitney *U*-test. ***$p < 0.001$. Lines represent the mean ± SD). **e**, **f** Individuals with high percentages of shorter pre-selection *TCRB* sequences present higher sharing indices (**e**) and lower Gini coefficients (clonality) (**f**) (Pearson's correlation). In **d**, **e** an outlier has been taken out of the statistical analyses

to substantially affect the chances of a TCRB passing through thymic selection, since the distributions before and after selection are quite similar (Fig. 5d, f; compare right and left panels).

We also found that pre-selection TCRBs in T1D patients display medium and long Jdels, medium Vdels and long D5dels and D3dels more frequently; this is associated with a lower frequency of short deletions, especially for the Jdel site (Fig. 5d, left panels). The insertion sites show the opposite patterns: higher frequencies of medium n1ins and long n2ins (Fig. 5e, f, left panels). Interestingly, for the post-selection TCRBs many of the differences are maintained, suggesting that these alterations survive through positive and negative selection (Fig. 5, right panels, and Supplementary Fig. 12). These results are not biased by HLA haplotypes, as T1D patients with the predisposing HLA haplotype *DRB1*03/DRB1*0401/DQB1*02/DQB1*03* are not outliers for any of the rearrangement events (Supplementary Fig. 12).

It has been shown in pre-selection repertoires that the deletion profiles vary from gene to gene, and that these profiles are highly conserved among healthy individuals[24]. This is suggestive of a conserved sequence-dependent nuclease activity[24]. Therefore, we next looked at *VDJ* gene usage in the pre- and post-selection repertoires (Supplementary Figs. 13–15). However, only one V gene (*TRBV7-2*) is under-represented in patients in the pre-selection repertoire, and therefore *VDJ* usage is not likely to be

the factor explaining the altered indel patterns. In fact, alterations in the deletion patterns in T1D patients (i.e., higher frequencies of clonotypes bearing long deletions) are found for most *V* and *J* genes, both for under-represented and equally used genes (Fig. 6). This is suggestive of an increased gene-independent exonuclease activity in patients during rearrangement.

In summary, we find that patients with T1D make abnormally fewer insertions and delete more nucleotides at *VDJ* recombination sites. These results imply that T1D is associated with major thymic rearrangement abnormalities that severely impact upon the characteristics of the peripheral repertoire.

**T1D TCRB rearrangement impacts upon self-reactive potential.** We next hypothesized that the abnormal thymic rearrangement processes that we observe in T1D patients might manifest as an extreme phenotype, namely the emergence of clonotypes that are exclusive to the disease setting and which are short, with abnormal indel patterns. We therefore defined T1D-enriched clonotypes (present in most patients and few healthy donors) and HD-enriched clonotypes, with the opposite presence pattern. We used cut-offs of presence in at least 7 out of 14 individuals (≥50%) from one group and at most 2/14 (≤14%) in another group, as this difference reaches $p < 0.05$ in Fisher's Exact test. This resulted in 2812 unique amino acid T1D-enriched sequences

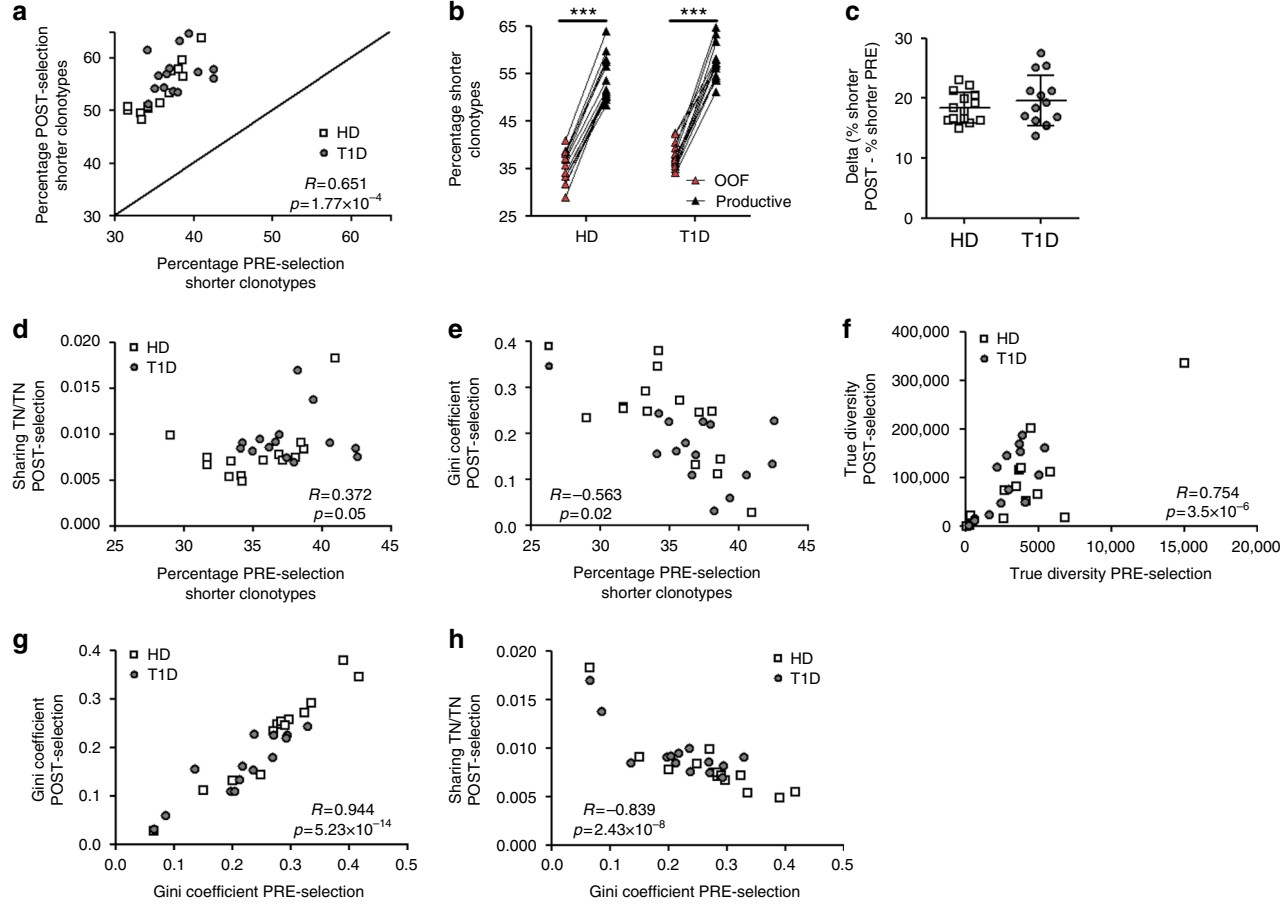

**Fig. 4** Correlations between diversity, sharing and TCRB CDR3 length pre- and post- selection. **a** TCRB CDR3 length correlates between pre- and post-selection repertoires. **b** Short TCRB CDR3s are enriched during thymic selection, both in type 1 diabetes patients (T1D) and healthy donors (HD) (paired t-test. ***$p < 0.001$). **c** There are no differences in the degree of enrichment in short TCRB CDR3s in T1D patients versus HD (unpaired t-test. Lines represent the mean ± SD). **d**, **e** The percentage of pre-selection shorter clonotypes correlates positively with the post-selection sharing (**d**) and negatively with the post-selection clonality (**e**). **f**, **g** Pre- and post-selection diversity indices correlate (**f**, true diversity; **g**, clonality). **h** Clonality of the pre-selection repertoire correlates with post-selection sharing. **a**, **d**–**h**: Pearson's correlation

and 3695 HD-enriched unique amino acid sequences. These clonotypes were found in TN cells (T1D-enriched: 60.31%. HD-enriched: 61.13%), CM cells (T1D-enriched: 33.50%. HD-enriched: 29.72%), Treg cells (T1D-enriched: 3.57%. HD-enriched: 4.45%) and Tscm cells (T1D-enriched: 2.65%. HD-enriched: 4.70%).

T1D-enriched clonotypes differed in several respects from their control counterparts. First, they showed higher frequencies (Fig. 7a), suggesting that they are more frequent in blood, and may have undergone more rounds of expansion (in the case of CM) or were common and seeded at higher frequency (in the case of TN). Second, T1D-enriched clonotypes showed differential usage of certain V families, V genes, and J genes (Fig. 7b–d). Third, they present an extreme phenotype regarding TCRB CDR3 length; T1D-enriched clonotypes are much shorter than HD-enriched (Fig. 7e), due to both a higher frequency of deletions and lower frequency of insertions (Fig. 7f, g); these differences are significant for each of the 6 rearrangement positions (Supplementary Fig. 16. Vdels: $p < 1 \times 10^{-14}$; Jdels: $p < 1 \times 10^{-73}$; D5dels: $p < 1 \times 10^{-19}$; D3dels: $p < 1 \times 10^{-04}$; n1ins: $p < 1 \times 10^{-19}$; n2ins: $p < 1 \times 10^{-08}$). These findings therefore recapitulate, to a more extreme degree, the aberrations seen in T1D pre- and post-selection TCR repertoires.

Finally, we compared amino acid usage in T1D-enriched versus HD-enriched clonotypes. T1D-enriched clonotypes show preferential amino acid usage in the central CDR3B positions (P5–P11)

(Supplementary Fig. 17); interestingly, amino acids that are over-represented in positions P6 and P7 include the hydrophobic residues proline, alanine, valine, methionine, tryptophan, phenylalanine and leucine. It has been reported that over-representation of hydrophobic residues at positions P6 and P7 of the TCRB CDR3 is a characteristic of self-reactivity[18], although these findings remain to be extended and their relevance clarified. Therefore, while the amino acid usage we observe in T1D-enriched clonotypes suggests a self-reactive bias, additional functional and structural analyses will be required.

In conclusion, we have shown that a disease-exclusive component of the T1D TCR repertoire manifests an extreme phenotype characterized by significant shortening of the CDR3B region (due to more deletions and fewer insertions), as well as enhanced interfacial hydrophobicity.

**Autoantigen-specific TCRB sequences are abnormally short.** To address the potential impact of short TCRB sequences on target specificity we examined whether TCRB chains from autoreactive T cells are shorter than virus-specific TCRs and fall within the shorter region of CDR3B length distributions obtained by deep sequencing (Fig. 2a–d).

For this analysis we used the same blood draw of 7/14 T1D patients and 6/14 healthy donors to isolate CD4+ T cells specific for the T1D autoantigen glutamic acid decarboxylase (GAD65)

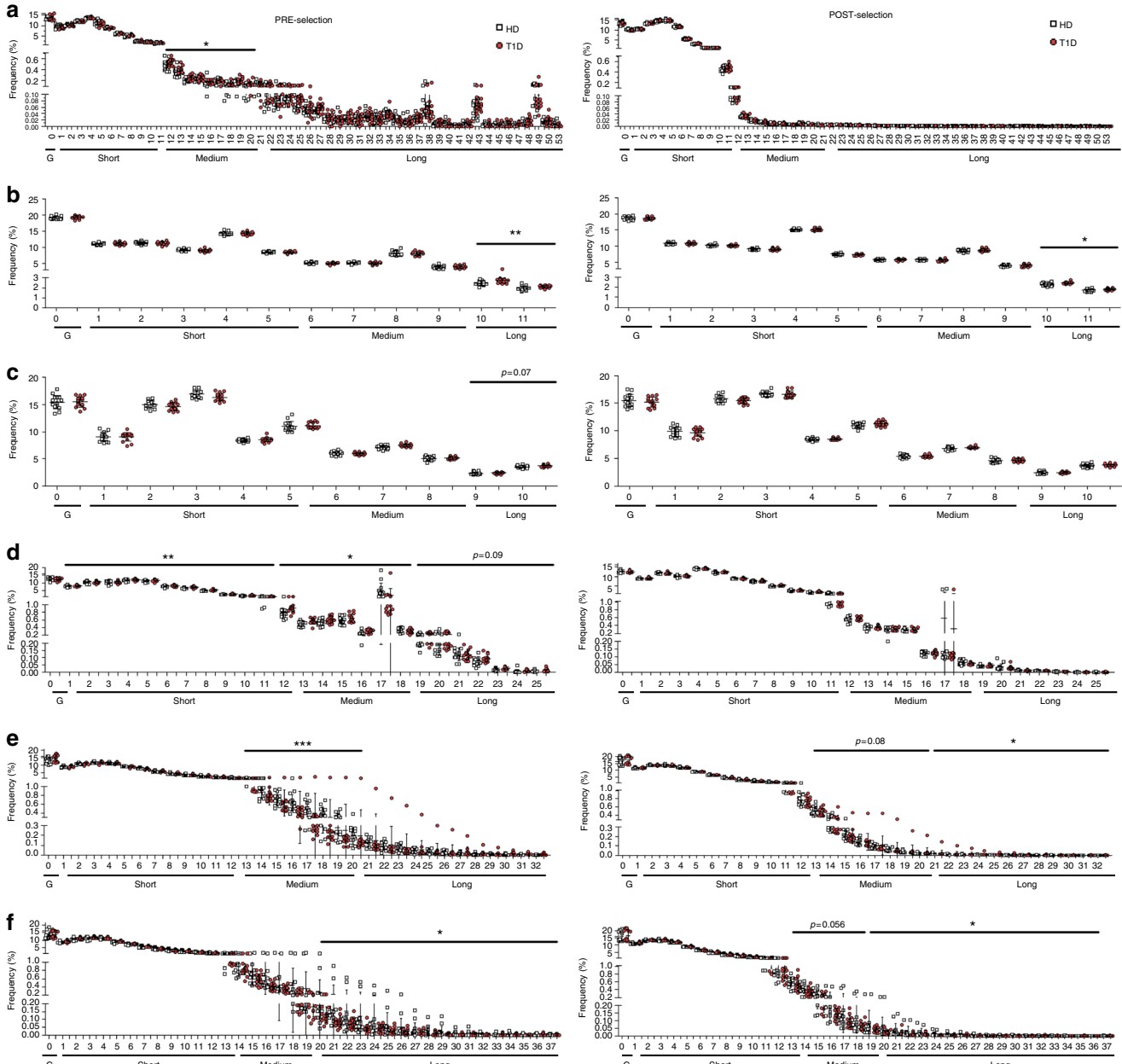

**Fig. 5** Type 1 diabetes patients present altered indel patterns during TCRB rearrangement. OOF unique sequences from true naive (TN) cells (pre-selection clonotypes) were analyzed for the number of deletions and insertions at each of the 6 rearrangement sites (left column). The same analysis was performed for productive unique sequences from the same TN cells (post-selection clonotypes) (right column). The deletions (and insertions) were classified as short, medium or long depending on their frequency in the overall distribution for each of the 6 rearrangement sites. **a** Vdel. **b** D5del. **c** D3del. **d** Jdel. **e** n1ins. **f** n2ins. G: germline (no deletions/insertions). Short: 1–11 (Vdel, Jdel), 1–5 (D5del, D3del), 1–12 (n1ins), 1–13 (n2ins). Medium: 12–21 (Vdel), 6–9 (D5del), 6–8 (D3del), 12–18 (Jdel), 13–20 (n1ins), 14–19 (n2ins). Long: 22–65 (Vdel), 10–11 (D5del), 9–10 (D3del), 19–25 (Jdel), 21–32 (n1ins), 20–37 (n2ins). Type 1 diabetes (T1D) patients display higher frequencies of long deletions and lower frequencies of long insertions. In **e** the outlier (T1D #2) was removed for the statistical analysis. In **d** the class 17 nucleotides was not included in the Medium range of Jdel, as this class behaves as an outlier when compared to the other Jdel classes. **a**–**f** Unpaired t-test or Mann–Whitney U-test. *$p < 0.05$. **$p < 0.01$. ***$p < 0.001$. Lines represent the mean ± SD

and for cytomegalovirus (CMV) after stimulation in vitro[26–28] and obtain *TCRB* sequences from single cells[29]. We identified 219 GAD65-specific *TCRB* sequences from 7 T1D patients, 155 GAD65-specific *TCRB* sequences from 6 healthy donors and 122 CMV-specific *TCRB* sequences from 1 control and 2 patients.

When all GAD65 clonotypes and all CMV clonotypes were pooled and their lengths compared, we found that GAD65 CDR3B length distribution shows a higher frequency of short clonotypes and a lower frequency of long ones compared with CMV (Fig. 8a). To examine where these antigen-specific

clonotype lengths lie in relation to representative TCR repertoires, their length distributions were compared with that of the global TCR repertoire obtained from deep sequencing of healthy donors (data from Fig. 2a–d). We found that, while CMV clonotypes follow the same length distribution as the global TCRs (Fig. 8b), GAD65 clonotypes are shorter (Fig. 8c).

We next aimed to validate these findings using autoreactive TCR repertoires published in T1D studies. We found that GAD65-specific clonotypes in T1D patients are shorter than in non-diabetic autoantibody-positive patients[30] ($n = 479$ and $n =$

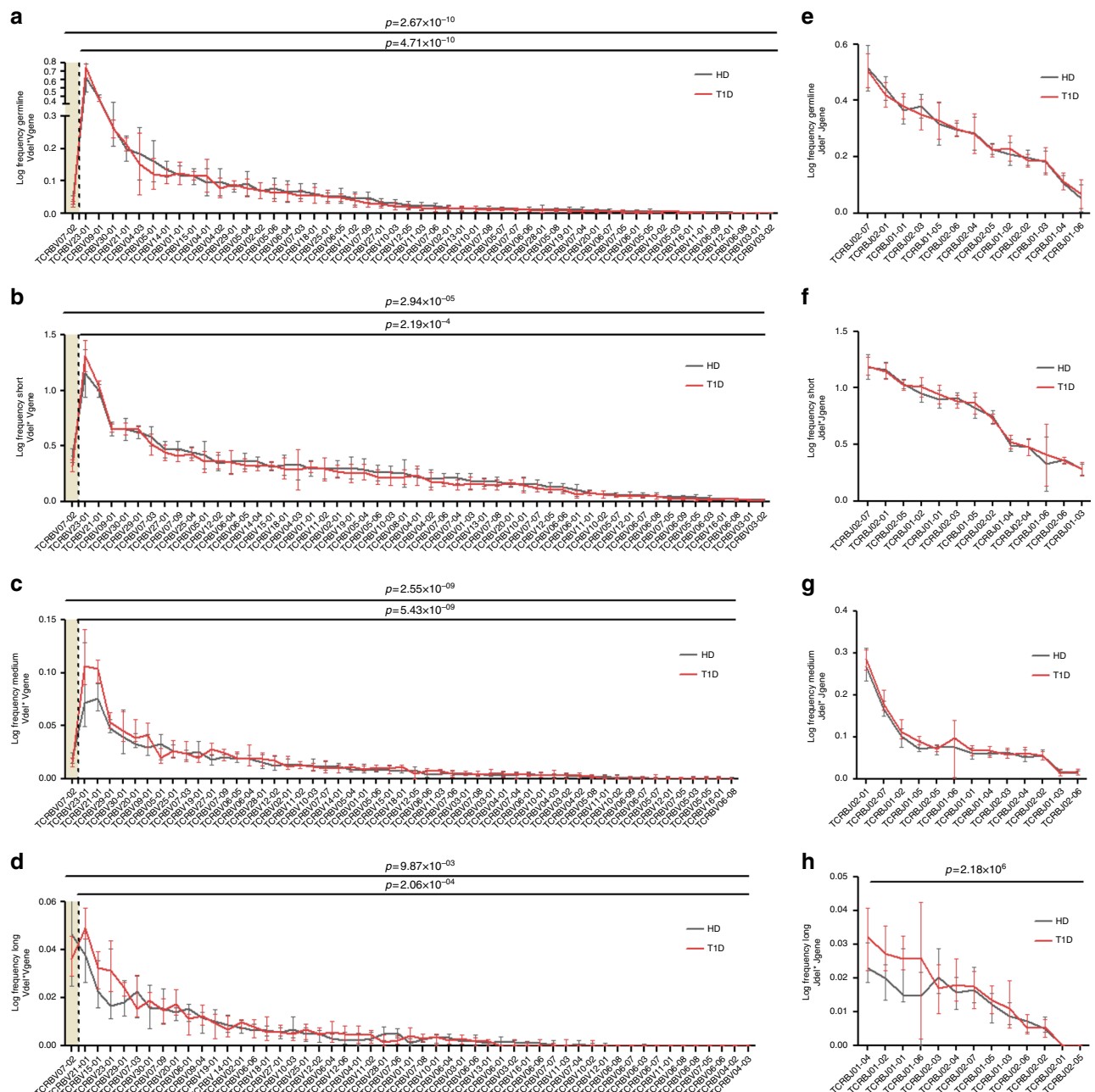

**Fig. 6** Alterations in deletion patterns in patients are found in most *V* and *J* genes. Frequency of germline **a**, **e**, short **b**, **f**, medium **c**, **g** or long **d**, **h** deletions for each of the *V* genes (**a**–**d**) and *J* genes (**e**–**h**) in type 1 diabetes (T1D) patients and healthy donors (HD). The gray shaded area represents *V* genes that are differentially used in patients versus controls (Supplementary Fig. 14). The increased deletions pattern of T1D patients is seen both in under-represented and equally used genes. The statistical test was performed for all genes or just for those equally used in T1D and healthy donors (i.e., without those genes included in the shaded areas). Mixed effects two-way ANOVA with individual as random variable. Lines represent the mean ± SD

387, respectively; Fig. 8d, compare red and blue lines) and again, these published T1D GAD65 clonotypes were shorter than the healthy donors' global TCR repertoire (Fig. 8d, compare red and black lines). Interestingly, the CDR3B length distribution of GAD65 clonotypes of non-diabetic autoantibody-positive patients[30] is not different to that of our healthy donor TCR repertoires (Fig. 8d, compare blue and black lines). Note that the CDR3B length distributions obtained from our own GAD65 single-cell data and GAD65 clonotypes from the literature[30] are almost identical (Fig. 8c, d, compare the shape of the red lines).

Finally, we also analyzed published TCR clonotypes from autoreactive CD4[+] T cells that have been identified by different methods in the T1D setting[31–38] (*n* = 39; Fig. 8e) as well as intra-

islet-proinsulin-specific CD4 T cell clonotypes[39] (*n* = 8; Fig. 8f) and compared these with the healthy donor TCR repertoire as before. We found that the CDR3B length distribution for proinsulin-specific[39] and autoreactive clones[31–38] shows a clear trend to being shorter than expected (Fig. 8e, f).

Summarizing these comparisons, autoantigen-specific TCRB clonotypes obtained after stimulation in vitro and from a variety of published methods show reduced TCRB CDR3 length when compared to both virus-specific clonotypes and a global TCRB length distribution obtained from healthy donors. These findings support the proposal that short TCR clonotypes present in patients with the autoimmune disease T1D are associated with autoreactivity.

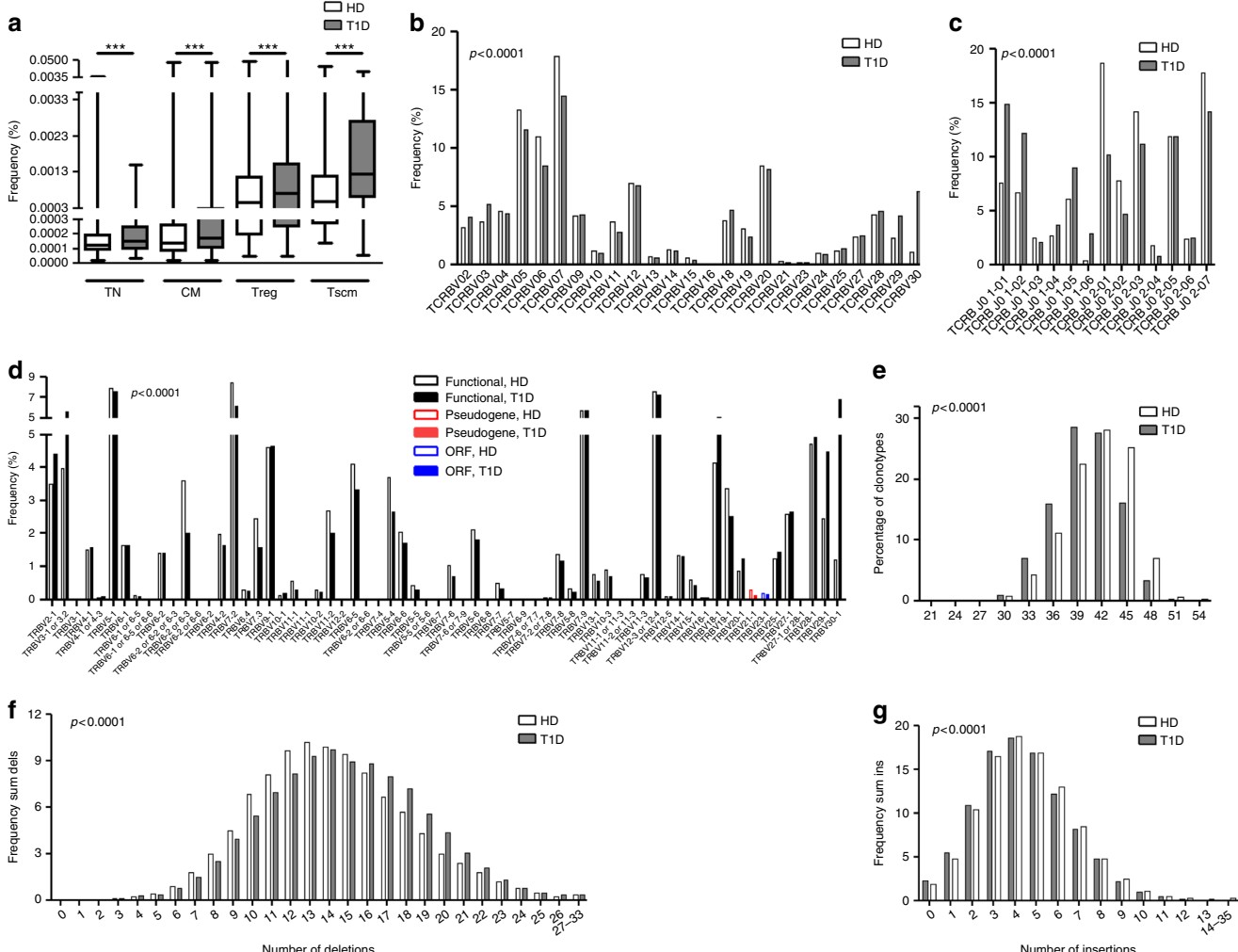

**Fig. 7** Type 1 diabetes-enriched clonotypes present an extreme phenotype. **a** The clonotypes belonging to the type 1 diabetes (T1D)-enriched group are more frequent than the healthy donor (HD)-enriched ones (Mann–Whitney U-test. ***$p < 0.001$. Lines represent the mean ± SD). **b**–**d** T1D-enriched clonotypes show differential usage of V families (**b**), J genes (**c**) and V genes (**d**). **e** T1D-enriched clonotypes are shorter than the HD-enriched ones. **f** T1D-enriched clonotypes have more nucleotides deleted (sum of Vdels, D5dels, D3dels and Jdels). **g** T1D-enriched clonotypes have fewer insertions (sum on n1ins and n2ins). **b**–**g**: $\chi^2$-test

## Discussion

There is a considerable gap in knowledge and understanding of the factors beyond those encoded in the germline that predispose to type 1 diabetes (T1D). Given the importance of autoreactive CD4+ T cells in initiating and perpetuating β-cell autoimmunity, we examined whether there are distinct, disease-associated features of the T cell receptor (TCR) in this setting, which could increase the risk of autoimmunity. We undertook a large examination of TCR repertoire characteristics in T1D to a degree not achieved hitherto. Most striking are our observations that the T1D TCR repertoire contains abnormally short β-chain CDR3 regions and that autoantigen-specific CD4+ T cells are enriched for shortened TCRB CDR3 sequences. We also find that TCR repertoires in T1D are abnormal in the degree of their diversity, publicity/sharing, and interfacial hydrophobicity. None of these abnormalities has been described previously. These features are not apparently human leukocyte antigen (HLA) or selection determined; rather they precede, and are independent of thymic selection events. Taken together these data suggest that early events in thymic T cell development and repertoire generation are abnormal, and have clear implications for a model of T1D risk that results from a generalized, heightened degree of self-recognition by T cells.

Our finding that pre- and post-selection TCRB repertoires are enriched of TCRBs bearing shorter CDR3 loops, arises from a combination of identifying more V/D/J deletions and fewer insertions during TCRB rearrangement. A model can be proposed to explain these findings (Fig. 9) in which random VDJ gene rearrangement typically yields a pre-selection repertoire of TCRB CDR3s across a range of lengths, but shorter lengths are favored in T1D. Following rearrangement, TCRBs bearing short CDR3s are enriched during positive selection[12–17,25] and our own data bear this out. As a consequence of these two biases operating in series, the outcome of positive selection in T1D is a richer and more diverse repertoire, and thus the pipeline into negative selection is already abnormal. There is evidence from elsewhere that negative selection events are dysregulated in T1D[40,41], causing failed deletion of autoreactive clonotypes. Thus, in our proposed model, the overall outcome is a more diverse post-selection repertoire, enriched in shorter clonotypes and highly shared among patients: a series of features that is borne out experimentally in the present study. Furthermore, our finding that autoreactive TCRB CDR3s are shorter than anti-viral and global, healthy donor repertoires suggests that diabetes-related enrichment in short TCRs augments opportunities for autoreactivity. It is also notable that some of these findings have been

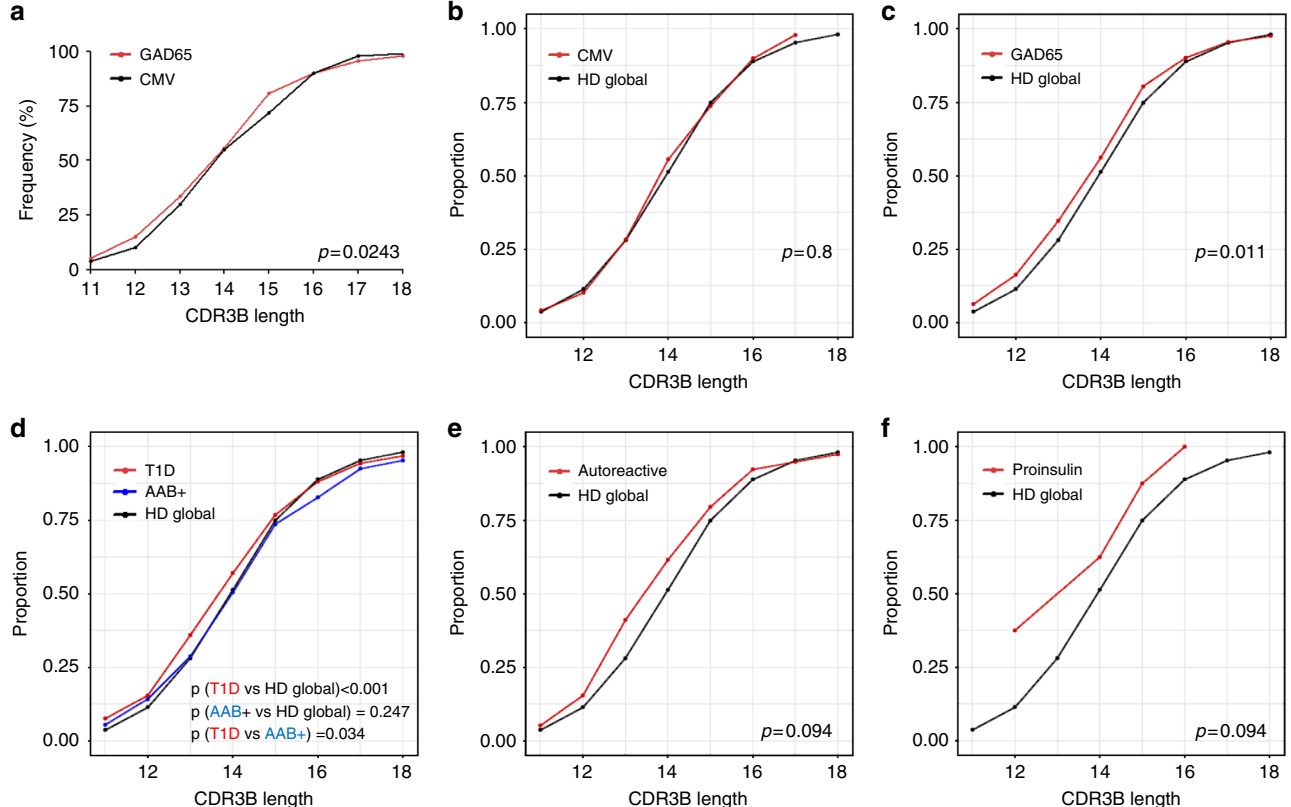

**Fig. 8** Autoreactive clonotypes show TCRB CDR3 length shortening. **a** Cumulative plots of glutamic acid decarboxylase 65 (GAD65)-specific ($n = 374$) versus cytomegalovirus (CMV)-specific ($n = 122$) TCRB clonotypes. **b** Cumulative plots of CMV-specific TCRB clonotypes versus healthy donor global TCR repertoire. **c** Cumulative plots of GAD65-specific TCRB clonotypes versus healthy donor global TCR repertoire. **d** Cumulative plots of GAD65-specific clonotypes of type 1 diabetes (T1D) patients from the literature ($n = 479$) versus GAD65-specific clonotypes of non-diabetic autoantibody-positive patients ($n = 387$) versus healthy donor global TCR repertoire. **e** Cumulative plots of T1D autoreactive clones described in the literature ($n = 39$) versus healthy donor global TCR repertoire. **f** Cumulative plots of intra-islet proinsulin-specific clonotypes ($n = 8$) versus healthy donor global TCR repertoire. **a–f**: One-sided bootstrapped Kolmogorov–Smirnov test

reported in patients with Rasmussen's encephalitis, an inflammatory disease of putative autoimmune origin[42], suggesting that these principles proposed by our model may be generalizable.

The mechanism through which more *V/D/J* deletions and fewer insertions arise is not known. Selection of *V*, *D* and *J* genes to be recombined is not stochastic; it depends upon recombination signal sequences[43] and on the accessibility and epigenetic modifications of the *TCR* loci[44–46]. Furthermore, there is evidence that mutations in the recombination-activating genes 1 (*RAG1*) and 2 (*RAG2*) generate TCR CDR3s with higher deletion rates, apparently by recruiting the alternative instead of the classical non-homologous end-joining pathway[47–49]. Hypomorphic polymorphisms in *RAG* are found in numerous immunodeficiency syndromes associated with autoimmune manifestations[50–52] and it was proposed that this might give rise to diverse TCR repertoires that predispose to autoimmunity[53]. Alterations in the insertion patterns can result from mutations in exon 9 of the *DNTT* gene encoding the active site of the terminal deoxynucleotidyl transferase (TdT)[54], although all T1D patients in the present study had the wild-type exon 9 sequence The balance of differentially spliced forms of TdT, which are known to differ in N-nucleotide addition and exonuclease activities[55], could also impact upon insertion patterns. Individual variation in the efficiency of the N-nucleotide addition process has also been described for B cell receptor (BCR) rearrangement[56]. Therefore, it remains possible that one, or a combination of several factors described previously, explain CDR3B length skewing. A systematic examination of the functional integrity of all of the

elements described, at all stages of T1D development, will probably be required to gain a better understanding of the underlying mechanisms of abnormal TCRB chain generation.

Several other observations in our study are of interest and could have bearing on the pathogenesis of T1D. True naive (TN) cells exhibit greater sharing, which may reflect a widespread impact of impaired TCR rearrangement and faulty negative selection[40,41]. A propensity to greater sharing in the T1D TN cell subset could be a consequence of the TCR rearrangement alterations we describe. It has been suggested that TCRs with fewer nucleotide insertions are more likely to be shared across individuals[23,57–60] and it is known that highly shared TCR pools are enriched in clonotypes bearing lower numbers of insertions[20].

HLA type did not appear to be a major confounder in our study, as the highest T1D risk alleles did not associate with excessive diversity, sharing or shortness, and our data point to pre-selection rearrangement processes, which are HLA independent, as being critical events. Nonetheless, numbers of individuals are small in these sub-analyses and the possibility that HLA type influences TCRB chain rearrangement will need to be clarified. A recent study has shown that usage of TCR *Vα* and *Vβ* genes is associated with the HLA genotype[61]; however, this effect was much stronger for *Vα* than *Vβ* genes, and *Vβ* genes whose peripheral usage was affected by HLA genotype are not preferentially enriched in our T1D cohort. In fact, the *V* gene usage is quite similar among patients and healthy donors. The only alteration found to be distinctive in T1D patients bearing *DRB1*03/DRB1*0401/DQB1*02/DQB1*03* is higher sequence sharing

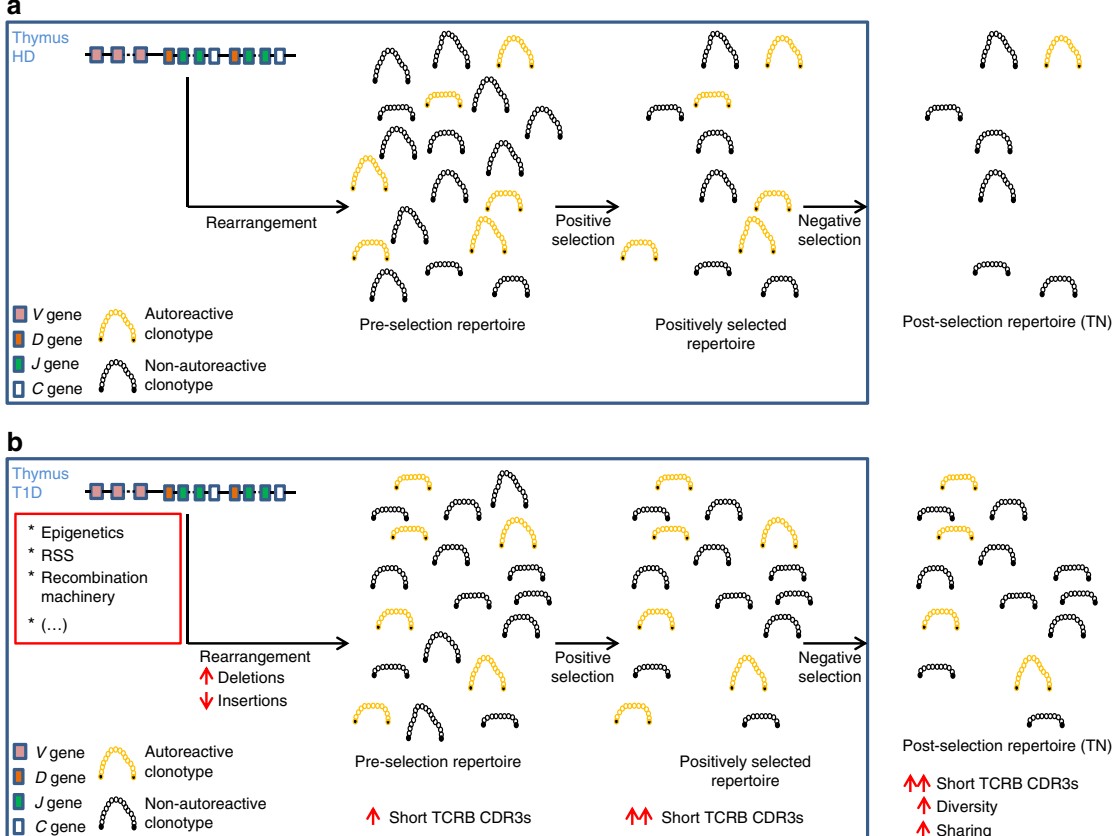

**Fig. 9** Proposed model of TCRB rearrangement and selection. **a** In healthy donors (HD) a random combination of *VDJ* genes undergoes rearrangement (see Supplementary Fig. 12 for a more detailed explanation of the rearrangement process), yielding a pre-selection repertoire of short and long TCRB CDR3s. Positive selection enriches for short TCRB CDR3s, and negative selection deletes most of the autoreactive clonotypes (orange), generating a polyreactive post-selection repertoire (true naive cells). **b** In type 1 diabetes (T1D) patients, alterations in TCRB rearrangement cause the generation of higher frequencies of shorter TCRB CDR3s, bearing long deletions and short insertions. Positive selection enriches for shorter TCRs, and therefore the input of TCRB chains into negative selection is higher than in healthy donors. Published alterations in negative selection suggest that in T1D a notable proportion of autoreactive clonotypes do not get deleted. The final outcome is a more diverse post-selection repertoire, enriched in shorter clonotypes, highly shared and containing autoreactive clonotypes. Black dots in clonotypes represent the conserved C and F amino acids in the 5′ and 3′ ends of the TCRB CDR3, respectively. White dots represent amino acids other than the conserved C and F. To ease the interpretation of the model only the CDR3B loop of the TCRB chain is shown, and the rearrangement of the TCRA chain has been omitted

among central memory cells (CM) and between the TN and CM compartments, which is consistent with a peripheral, rather than central effect. To the extent that we could, other potential major study confounders were addressed, such as age matching, since ageing is related to thymic involution[62] and decreased TCR repertoire diversity[63].

To gain a greater insight into disease-associated repertoire effects, we defined T1D-enriched and HD-enriched clonotypes. T1D-enriched clonotypes have interesting properties, akin to an extreme phenotype: they are present at higher frequencies, are short, and bear marked interfacial hydrophobicity, a feature associated with self-reactive mouse and human TCRB repertoires[18] and which requires further examination in additional settings.

High-throughput *TCRB* sequencing can be performed using genomic DNA (gDNA) or cDNA synthesized from mRNA. We elected to use cDNA in this study because we wished to focus on expressed receptor transcripts, representing the circulating, *in vivo* repertoire, and explore how these may deviate from normal during an active autoimmune process. In our preliminary studies (see Methods), TCRB repertoires showed a very high degree of overlap when acquired from cDNA (rearranged and expressed receptors) and gDNA (rearranged receptors, expression

not known), suggesting that the approaches may indeed provide comparable results. The number of TCR templates available for sequencing when using gDNA is lower than when using RNA, giving the latter a broader repertoire coverage from a given amount of starting template[64]. Future studies focused on direct cDNA versus gDNA comparisons in large sample sets will be required to establish the optimal approach for each different study question.

Use of cDNA rather than gDNA is potentially susceptible to the influence of TCR gene expression level per cell, and particularly to the potential bias caused by rare populations of highly activated T cells, which may differ in the setting of an active autoimmune disease. We saw no evidence that T1D patients and healthy donors differed in activation levels or in CD3 expression, even at the 99.9th percentile (see Methods). Although the possibility remains that a very small number of high TCR-expressing cells influence some of our results, we observed a strong correlation between number of cells analyzed and the number of unique clonotypes sequenced, arguing against over-representation of TCRs due to high RNA expression levels. Furthermore, we avoided bias in calculating true diversity and clonality indices by using normalization to cell number, supported by our observation that cell number and number of

unique reads are highly correlated. In addition, most of our analyses (including CDR3 length distribution, *VDJ* gene usage, overlap calculation, indel analysis, correlations and T1D-enriched versus HD-enriched analyses) were carried out on unique sequences, avoiding biases due to differences in sequence frequencies. A final consideration in relation to technical biases that could influence measurement of CDR3B length distributions is that PCR might be more efficient for shorter templates. However, all PCR and sequencing reactions for HD and T1D patients were conducted together and corrected simultaneously with the same approach to avoid systematic error, so any potential for PCR skewing toward the detection of shorter lengths would apply equally to both T1D patient and healthy donor groups.

Future studies should also aim to examine the generalizability of our findings (e.g., CD8[+] T cell repertoire; other autoimmune diseases; pre-T1D; rate of progression; age of T1D onset), as well as gain a better understanding of the molecular mechanisms of abnormal TCR rearrangement. In this respect, the outlier patient with extreme insertion and Jdel patterns, although having an unremarkable clinical history, warrants greater scrutiny of T cell autoreactivity.

In summary, we show for the first time extensive abnormalities in TCRB rearrangement processes in an autoimmune disease setting, findings consistent with the contention that antigen–receptor repertoire should be included as a major contributor to autoreactivity and thus disease risk.

## Methods

**Human subjects**. 14 patients with new onset T1D and 14 healthy donors were recruited into this study, which was approved by the South London Research Ethics Committee 5, REC reference 08/H0805/14. All recruited patients and healthy donors provided written informed consent. In the case of Treg and Tscm cells only cells from 8 of the patients and 8 of the healthy donors were isolated. Patients and healthy donors were typed for *DRB1\** and *DQB1\** alleles by the Tissue Typing Service at Guy's Hospital (London, UK).

**Cell staining and sorting**. An average of 116 mL of blood was obtained from a single draw from volunteer into heparinised Vacuette tubes (Greiner Bio-one, Gloucestershire UK). PBMCs were immediately isolated by density gradient centrifugation using Lymphoprep (Axis-Shield PoC AS, Oslo, Norway), and a mean of $94 \times 10^6$ cells were directly stained with the following monoclonal antibodies (indicated in brackets is clone name, catalog number and volume in μL used to stain $35 \times 10^6$ PBMCs—for higher cell numbers volumes were increased accordingly): anti-CD14 (HIB19, 561121, 5 μL), anti-CD19, (M5E2, 561391, 5 μL), anti-CD3 (SK7, 641415, 62.5 μL), anti-CD4 (SK3, 345768, 25 μL), anti-CD8 (RPA-T8, 562311, 2.5 μL), anti-CD25 (M-A251, 555432, 200 μL), anti-CD45-RO (UCHL1, 337168, 20 μL), anti-CCR7 (150503, 562555, 62.5 μL), anti-CD95 (DX2, 561978, 25 μL) BD Biosciences; anti-CD127 (eBioRDR5, 45-1278-42, 50 μL) eBiosciences; anti-CD27 (O323, 302830, 10 μL) Biolegend; and LIVE/DEAD Fixable Aqua Dead Cell Stain (Thermofisher). Flow cytometric sorting took place in a BD FACSAria II flow cytometer (Becton Dickinson, San Jose, CA, USA); the fluids system was flushed before sorting with RNASe Zap (Thermofisher). Events were gated as shown in Supplementary Fig. 1, and TN, CM, Treg and Tscm cells sorted with a flow rate of 2000–5000 events/s into DNase/RNase-free Eppendorf tubes containing 300 μL of RNase-free PBS, using a 70 μM nozzle. The cells were centrifuged at 1500xg 10 min at room temperature, the supernatant discarded and the cell pellet lysed with 350 μL of RLT buffer (QIAGEN). Lysates were stored at −80 °C until further processing. All sorted populations were greater than 98% pure. Samples were always stained and sorted in pairs (e.g., one healthy donor and one T1D patient per experiment) to minimize batch effects.

**RNA isolation and cDNA synthesis**. RNA was isolated using the RNeasy Micro kit (for samples with < 650,000 cells) or RNeasy Mini kit (for all other samples) (QIAGEN. 74004 and 74106, respectively), following the manufacturer's guidelines, including treatment with DNase (RNase-Free DNase Set, QIAGEN, 79254) to avoid contamination with genomic DNA. RNA was eluted twice with 14 μL (Micro kit) or 30 μL (Mini kit) of RNase-free water, and quantified using a NanoDrop 1000 Spectrophotometer (Thermo Scientific). Isolated RNA was used on its entirety for cDNA synthesis using the High Capacity cDNA Reverse Transcription Kit (Applied Biosystems, 4368814). Samples from healthy donors and T1D patients were processed together to avoid batch effects, and each cell subset (TN, CM, Treg, and Tscm) processed separately to impede cross-contamination.

**Sequencing of TCRB repertoires**. cDNA was sent to Adaptive Biotechnologies, Corp., (Seattle, USA) for deep (TN, CM, and Treg cells) or survey (Tscm) sequencing of *TCRB* sequences using the ImmunoSEQ assay[23,65,66]: a multiplex PCR system is used to amplify *TCRB* sequences from the whole cDNA sample; this approach generates 87-base-pair fragments capable of identifying the VDJ region spanning each unique CDR3B. Amplicons are then sequenced using the Illumina HiSeq platform. Using a baseline developed from a suite of synthetic templates, computational corrections were used to correct for potential PCR bias[65]. Low frequency reads were binned and both PCR and sequencing errors were removed as described[23,65,66]. Briefly, a nearest-neighbor algorithm based on Hamming distance was used to collapse the data into unique sequences by merging closely related sequences (Hamming distance ≤ 2). Singletons that do not have a consensus sequence that they can merge into were considered sequencing errors and discarded. Reads were compared against the IMGT database (www.imgt.org), yielding for each TCRB read V, D, and J gene usage, CDR3B length (i.e., number of nucleotides from the codon coding for the second conserved Cys in the V gene to the codon coding for the conserved phenylalanine in the J gene), and number of nucleotides deleted/inserted in each of the 6 rearrangement sites (Vdel, D5del, D3del, Jdel, n1ins and n2ins). A mean of 3,568,713 (range 1,006,045–12,954,066) and 1,008,767 (range 254,789–3,580,864) total reads were obtained for deep and survey sequenced samples, respectively.

**Bioinformatic analyses**. Analyses shown in this study were performed using Adaptive Biotechnology's ImmunoSEQ online tool, the R package tcR[67], KNIME, GraphPad Prism, IBM SPSS Statistics 22, and in-house R scripts.

**Comparison of cDNA and gDNA as nucleic acid source**. To investigate the appropriateness of RNA (and hence cDNA) for studying the TCRB repertoires shown in this paper, we sorted TN and CM cells from 3 healthy donors (HD#15, HD#16 and HD#17), to compare TCRB repertoires obtained from cDNA and genomic DNA (gDNA) from the same sorted samples.

A mean of 3,232,311 TN cells (range 3,137,083–3,409,198) and 1,478,741 CM cells (range 698,153–2,138,070) were sorted as described in Material and Methods section. RNA and gDNA were isolated simultaneously from each of the samples using the AllPrep DNA/RNA/Protein Mini Kit (QIAGEN, 80004). cDNA synthesis and TCRB sequencing of both cDNA and gDNA material for each sample were carried out as described in the Material and Methods section. Repertoire overlap between cDNA and gDNA is very high (Supplementary Fig. 18).

Since the use of cDNA has the potential to over-represent rare populations of highly activated T cells we examined our samples for evidence of differences in activation levels between patients and controls. There was no difference in average activation level of CM and TN cells (as measured by surface CD25 expression; Supplementary Fig. 19a, b), or in the frequency of CD25[+] cells (CM cells: HD, 40.08% ± 10.08%; T1D, 41.02% ± 6.37%. TN cells: HD, 4.84% ± 2.80%; T1D, 6.33% ± 2.52%. $p = $ NS. Student's *t*-test). A more focused analysis of the top 2% of highly activated cells (CD25high) also demonstrated no difference in activation levels (Supplementary Fig. 19c, d), including at the 99.9[th] centile. The top 2% and 99.9th centile of highly activated cells also did not differ for CD3 expression (as a surrogate for TCR expression. Supplementary Fig. 19e, f). Thus in this sample set, with this approach, we did not find evidence of a difference in highly activated populations in the circulation in patients compared with healthy donors. As a result of these preliminary studies, we elected to use cDNA for sequencing in order to assess the repertoire of expressed (and not only rearranged) receptors.

**Calculation of diversity indices**. Diversity indices are highly influenced by differences in sample size. To control for differences in sample size and depth of sequencing the count of each clonotype was normalized to the cell number as follows:

$$\text{Normalized count}_{i,y} = \frac{\text{Count}_i}{\text{Cell number}_y} \times \text{cell number of sample with lowest cell number}$$

(1)

where $i$ is a particular clonotype appearing in sample $y$. This normalization of counts took place separately for each of the four cell subsets; therefore, the normalization factor cell number of sample with lowest cell number was $1 \times 10^6$ for TN, 363,555 for CM, 210,070 for Treg and 52,626 for Tscm samples. A detection limit for rare clonotypes was defined as the normalized count of the rarest clonotype in the sample with the lowest cell number among all samples of the given cell type; all clonotypes with normalized counts lower than the detection limit were excluded from calculations of diversity measures. True diversity and Gini coefficient (a measure of clonality) were calculated using the tcR package[67]. Both parameters were calculated as they provide complementary information about the overall diversity of one given sample.

**Calculation of overlap indices**. Sharing among TCR repertoires was quantified by calculating the overlap coefficient (overlap (X, Y) = |X and Y|/min (|X|, |Y|) for nucleotide sequences (species = nucleotide sequence).

**Correlations analyses**. Correlations were calculated using Pearson correlation (for parametric variables) or Spearman (for non-parametric variables). Some new variables were created to be able to calculate meaningful correlations within and among pre- and post-selection repertoires. Sharing cell type 1/cell type 2 refers to the mean cell type 1 to cell type 2 overlap coefficients of a given individual with all other individuals. To be able to correlate frequency of appearance of short TCRB CDR3s with other variables we calculated the cumulative frequency of post-selection TCRB CDR3s of 27–42 nucleotides long (percentage of shorter post-selection clonotypes), and the cumulative frequency of pre-selection TCRB CDR3s of 14–41 nucleotides long (percentage of shorter pre-selection clonotypes).

**Analysis of insertion and deletion patterns**. For each of the 6 rearrangement sites (Vdel, D5del, D3del, Jdel, n1ins, and n2ins) the frequency of clonotypes bearing a particular number of indels was calculated for TN cells of each of the 28 individuals (the example below relates to post-selection Vdels; the same approach was followed for all 6 rearrangement sites and for pre- and post-selection repertoires):

$$\text{Frequency of post} - \text{selection } i \text{ Vdels} = \frac{n \text{ of productive unique clonotypes with } i \text{ Vdels}}{\text{total number of productive unique clonotypes}} \times 100$$

(2)

where $i$ refers to a particular number of Vdels. In the case of pre-selection analyses the same formula was employed, using OOF sequences instead of productive sequences.

In some cases the indel patterns are shown classified into germline, short, medium and long. Indels were classified as follows (the numbers refer to the length in nucleotides of the relevant rearrangement event): germline (no deletions/insertions); short: 1–11 (Vdel, Jdel), 1–5 (D5del, D3del), 1–12 (n1ins), 1–13 (n2ins); medium: 12–21 (Vdel), 6–9 (D5del), 6–8 (D3del), 12–18 (Jdel), 13–20 (n1ins), 14–19 (n2ins); long: 22–65 (Vdel), 10–11 (D5del), 9–10 (D3del), 19–25 (Jdel), 21–32 (n1ins), 20–37 (n2ins). A cumulative frequency was then calculated for each of the 4 classes and the 6 rearrangement sites, both for pre- and post-selection repertoires. Which lengths to include in each class (for each of the 6 rearrangement sites) was decided by examining the frequency distribution of all possible lengths in Fig. 5. In the case of n1ins the outlier highlighted in green in Supplementary Fig. 12 was removed for statistical analyses (see comments in Discussion). In the case of Jdel the class 17 nucleotides was not included in the calculation described above for Medium Jdels as this class behaves as an outlier when compared to the other Jdel classes (Fig. 5d), and would bias the interpretation of the statistical analyses.

To analyze which variables affect the frequency of Vdels we performed, for the pre-selection repertoire (i.e., TN out of frame sequences), a mixed effects two-way analysis of variance for the frequency of Vdel, with type of individual (T1D patient or healthy donor) and $V$ gene as fixed factors and individual as random variable (full factorial model). The frequencies of Vdels were calculated for each of the 28 individuals and each of the 53 $V$ genes as follows:

$$\text{Frequency of } i \text{ Vdels for TCRBVX} = \text{Log}_{10}\left(\left(\frac{n \text{ of TCRBVX unique clonotypes with } i \text{Vdels}}{\text{total number of unique clonotypes}}\right) \times 100 + 1\right)$$

(3)

where $i$ refers to a particular number of Vdels (range 0–65), and TCRBVX refers to one of the 53 $V$ genes included in the analysis. The analysis was carried out for all $V$ genes or just for the $V$ genes equally used between T1D patients and healthy donors, to account for potential interference of differentially used $V$ genes in the model (gray shaded areas in Fig. 6a). The same approach was used to investigate which variables affect the frequency of Jdels.

**Amino acid usage in T1D- and HD-enriched clonotypes**. Analysis of amino acid usage in T1D-enriched versus HD-enriched clonotypes was performed using Ice-Logo[68]. IceLogo builds on probability theory to visualize significant conserved sequence patterns in multiple peptide sequence alignments against background (reference) sequence sets that can be tailored to the system studied and protocol used. T1D-enriched clonotypes were compared against HD-enriched clonotypes (negative reference set) for each of the more frequent TCRB CDR3 lengths (12–15 amino acids long). All settings were left as default, and fold change was chosen as the scoring system. The IceLogo plots show those amino acids whose usage is significantly different (i.e., $p < 0.05$) between T1D-enriched and HD-enriched clonotypes.

**Isolation of antigen-specific CD4 T cells and single-cell PCR**. We isolated antigen-specific CD4+ T cells using an assay previously developed by Roederer and colleagues[26,27] and adapted by us for this purpose[28]. Briefly, from the same blood draw of 7 of the described 14 T1D patients and 6 of the described 14 healthy donors, fresh PBMCs ($2 \times 10^7$ per antigen condition) were incubated at $10^6$ cells/mL in 48-well plates (37 °C, 5% $CO_2$, 1 mL/well) in RPMI 1640 with Glutamax supplemented with penicillin/streptomycin, Amphotericin B (Fisher Scientific, 15140 and 15290) and 10% human AB serum (Sigma, H4522) containing 2 µg/mL of anti-CD40 monoclonal antibody (Biolegend, clone G28.5, 303611) and 10 µg/mL

of either GAD65 protein (Dyamid Diagnostics, 10-65702-15-01), CMV grade 2 antigen (containing antigens from all parts of the virus cycle of replication, Microbix Biosystems Inc, EL-01-02). As a positive control $2 \times 10^6$ PBMCs were incubated as above with 10 µg/mL of the polyclonal stimulus Staphylococcal enterotoxin B (SEB, Sigma-Aldrich, S4881). The negative control consisted on $2 \times 10^6$ PBMCs incubated as above with no additional stimulus. Non-adherent cells were harvested after 18 h, washed, and stained with the following fluorochrome-conjugated monoclonal antibodies (indicated in brackets is clone name, catalog number and volume in µL used to stain $2 \times 10^6$ PBMCs—for higher cell numbers volumes were increased accordingly): anti-CD14 (TuK4, MHCD1428, 2 µL), anti-CD19 (SJ25-C1, MHCD1928, 2 µL) (Invitrogen); anti-CD3 (SK7, 641415, 2 µL), anti-CD154 (TRAP1, 555700, 2.5 µL), anti-CD69 (FN50, 555530, 2.5 µL) (all from BD Biosciences); anti-CD4 (SK3, 46-0047-42, 3 µL. eBiosciences); and LIVE/DEAD Fixable ViVid Dead Cell Stain (Molecular Probes, L34955).

Flow cytometric sorting took place in a BD FACSAria II flow cytometer (Becton Dickinson, San Jose CA, USA) with FACSDiva software version 7.0. The fluids system was flushed before sorting with RNAse Zap (Thermofisher, AM9780). FSC and SSC doublets were excluded, and live CD3 + lymphocytes were defined as CD19− CD14− CD3+ ViViD−. Live CD3+ lymphocytes were further gated by FSC/SSC, and CD4+ cells gated. Antigen-specific CD4+ T cells were further defined as CD154+ CD69+[26–28], and sorted with a flow rate of 2000–5000 events/s into 96-well PCR plates (Thermofisher) containing 5 µL of RNAse/DNase-free PBS, using a 70 µM nozzle. Plates were snap frozen in dry ice and stored at −80 °C until processing.

Single-cell PCR was performed as described previously[29]. Briefly, cDNA was synthesized directly from single cells using 10 µL of qScript cDNA Supermix (Quanta Biosciences, 95048-025). A first round of PCR was done on total cDNA using TATAA GrandMaster Mix (TATAA Biocenter, TA05-50) in the presence of a mix of forward tagged variable region primers covering all Vβ genes and a reverse constant TCRB region primer (Supplementary Table 2) with the following conditions: 1 cycle of 95C 8′; 20 cycles of 95 °C 45″, 49C 1′ (with an increase of 0.3 °C per cycle), 72C 1′30″; 1 cycle of 72C 7′. A seminested second round PCR was performed on 3 µL of the first round product, using the tag primer (HTSP oligo) and a nested constant region primer (Supplementary Table 2), with Takara ExTaq DNA polymerase (Takara BIO INC, RR001A), with the following conditions:1 cycle of 94C 1′; 40 cycles of 56C 40″, 72C 30″; 1 cycle of 72C 6′. Each single-cell PCR product was Sanger sequenced using the HTSP oligo, and the sequencing results referenced to the IMGT database[69], to obtain V, D, and J gene usage, CDR3B length, and number of nucleotides deleted/inserted in each of the 6 rearrangement sites (Vdel, D5del, D3del, Jdel, n1ins, and n2ins).

**Statistical analysis**. For comparisons of two groups, paired or unpaired two-tailed t-student t-test (for normally distributed variables) or Mann–Whitney U-test (for other variables) were used. When pertinent, Benjamini-Hochberg multiple comparisons correction (false discovery rate—FDR—of 0.05) was applied. Pearson's $\chi^2$-t-test or Fisher's exact test were used for comparing groups in analysis involving qualitative variables. One-sided bootstrap version of univariate Kolmogorov–Smirnov test was used for variable distribution comparisons. Unless otherwise stated, bars represent means and error bars standard deviations. In box-and-whisker plots, the box extends from the 25th to 75th percentiles, line represents the median and whiskers the minimum and maximum values.

**Subsampling method**. To test whether observed differences (such as those in TCRB CDR3 length) are not an artifact of sub-sampling of TN cells and variable number of reads among samples, we performed binomial sampling of the number of sequences in the least abundant sample for each sample, for TN cells. This was repeated 500 times and TCRB CDR3 length distributions were computed for each replication. Frequencies of occurrence of each length were compared between T1D patients and healthy donors with Mann–Whitney U-test.

**Data availability**. All sequencing data have been deposited and made public in the Open Science Framework database (DOI 10.17605/OSF.IO/YDGTV)[70]: http://osf.io/ydgtv/. All other data are available from the authors upon reasonable requests.

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

## Acknowledgements

The study was supported by the National Institute for Health Research (NIHR) Biomedical Research Centre based at Guy's and St Thomas' National Health Service (NHS) Foundation Trust and King's College London (IS-BRC-1215-20006). I.G.-T. was supported, sequentially, by a postdoctoral fellowship from Fundación Barrié de la Maza, and by a Marie Curie Intra-European Fellowship (PIEF-GA-2012-327908). We are grateful to study volunteers for their participation and to Laura Eckhardt and Dr Jake Powrie for assistance in participants recruitment. We are grateful to Professor Jo Spencer for her assistance in this project.

## Author contributions

I.G.-T. designed and performed experiments, analyzed data and wrote the manuscript; A. L. analyzed data; I.G.-T., A.L. and M.P. conceived ideas and oversaw the research. Y.K. and R.B. performed experiments. M.P. is the guarantor of this work and, as such, had full access to all of the data in the study and takes responsibility for the integrity of the data and the accuracy of the data analysis.

## Additional information

**Competing interests:** The authors declare no competing financial interests.

