## [Peer Review File · Nature Communications]

Reviewers' comments:

Reviewer #1 (Remarks to the Author):

The manuscript by Gomez-Tourino proposes global alterations in TCR gene usage and recombination events from defined T cell subsets in subjects with type 1 diabetes. Overall, the manuscript is well designed and presented. The cohorts are well matched in terms of subject demographics and the T cell subsets are well defined and highly pure following FACS enrichment.

The main concern with the current conclusions is that the methods are analyzing TCR gene usage from cDNA rather than gDNA of sorted cell subsets. This practice is then susceptible to alterations in TCR gene expression per cell, with large variance possible from quiescent T cells to those of highly activated subsets that could dramatically skew the results and data interpretation. This is particularly important when comparing T cells from autoimmune donors to those of normal healthy controls. The authors should clarify and/or provide some data to clearly demonstrate that the results they have observed are in fact a general phenomenon observable at the gDNA level from sorted T cell subsets where one read is equivalent to one clone.

In addition to the concern above, there were other minor issues that should be addressed.

-The discussion of hydrophobicity at specific residues (positions P6/P7) promoting autoreactivity is largely extrapolated from animal model data or from limited autoreactive human T cell clones. Therefore, the discussion of this topic should be somewhat more guarded than what is currently presented that suggests this finding is a universally accepted phenomenon.

-While the number of subjects is small and may preclude analysis at this time, the authors should discuss what genetic risk variants, epigenetic processes, and/or developmental processes may account for this proposed universal skewing of the repertoire observed. Is this phenomenon observed in at-risk double autoantibody positive subjects prior to overt disease? Can this TCR analysis discriminate type 1 diabetes subjects from controls?

-What are the statistical tests that were run for the conclusions that are stated in Figure 2?

-The type 1 diabetes cohort appeared to have a number of older individuals. Is this phenomenon also observed in young type 1 diabetes donors that tend to exhibit a more rapid progression of the disease and loss of residual C-peptide?

-Table 1, limit to appropriate number of significant figures.

-Correct typo in legend of Figure 1

-Figure 2 scale bar and text is illegible in its current form. In addition, the panels could be more clearly labeled with cohort designations and AA or nucleotide information. Clarify statistical tests for e-h panels and display data from the whole cohort summarized.

-Figure 3 stats should be clarified for b-c

Reviewer #2 (Remarks to the Author):

In the manuscript by Dr. Peakman and colleagues, the authors are trying to demonstrate that

structural features of the TCR repertoire confers risk of autoimmune diseases. To address this hypothesis, they did an extensive analysis of TCR sequencing with a focus on TCRB rearrangements of 4 major CD4+ T cell subsets from patients with type 1 diabetes and matched healthy donors. Their main results are that TCRB sequences show greater diversity in type 1 diabetes patients and that TCR β chain sequences are short and highly shared across type 1 diabetes patients. Detailed analyses support that type 1 diabetes patients display peripheral TCR repertoire abnormalities that are present in the pre-selection TCRB repertoire and that these alterations are perpetuated into the post-selection of CD4+ T lymphocytes. Finally, they show that patients with type 1 diabetes delete more nucleotides at recombination sites during TCRB rearrangement. Taken together their work led to demonstrate that early events in thymic rearrangement during the CD4+ lymphocyte development impact upon the peripheral repertoire and this could be the basis of the risk of developing an autoimmune disease.

General comment

The authors provide a very huge amount of data on the repertoire of TCRB. The results are potentially interesting as they attempt to show that the alteration in TCRB repertoire in CD4+ T cell in type 1 diabetes patients is present in early event of CD4+ thymocyte maturation. However, the presented data do not fully gain a new insight into disease associated repertoire effects but they instead represent a collection of interesting observation in the generation of TCRB repertoire.

Specific major points:

- It is no clear to me if these clonotypes with shorter CDR3 length interact MHC/peptide complex and if they have anything to do with the immunopathology. In that respect, no data demonstrates that the abnormal TCRB repertoire influence the self recognition by T cells and if the CD4+T cell bearing these TCRB chain with shorter CDR3 loops interact more efficiently or less with MHC/peptide.
- The authors try to explain the mechanism through which more V/D/J deletion might arise. If, as stated by the authors, the presence of TCR β chain with shorter CDR3 loops is due to an alteration of VDJ recombination complex, this should be found in whole lymphocyte receptors chains. Therefore, it would also be interesting to check if the CD8+T cells TCR β chains exhibit the same defects.
- Somehow these results are not original, many publications are already reported that type 1 diabetes patients display a disturbed T cell repertoire and as mentioned by the authors both shortening of TCRB CDR3 length and sharing clonotypes are already been observed in patients with others autoimmune diseases.

Reviewer #3 (Remarks to the Author):

In this manuscript, Peakman and colleagues performed TCR repertoire sequencing to characterize the circulating CD4+ T cell populations from eight type 1 diabetes (T1D) patients. T1D is a well-known CD4+ T cell-mediated autoimmune disease and its etiology has been believed to be rooted in the abnormal thymic selection. In this paper, the authors indeed identified several abnormal features in these eight patients for their TCR CDR3 region, such as inter-subject sharing, shorter length, and over-representation of hydrophobic amino acids. Most importantly, by analyzing out-of-frame sequences of naïve T cells as surrogates for pre-selection TCR repertoire, the authors recapitulated all these abnormal features. This is a truly novel and significant discovery that, rather than thymic selection, the autoimmunity of T1D is rooted in the VDJ recombination step. However, there are concerns on the substance to support this conclusion:

General concerns:

1. Sample size. All major conclusions were made based on a rather small sample size. The previous experience in the field is that many abnormal features identified in the TCR repertoire, such as length

difference, V/J usage difference, or amino acid bias, disappeared when tested sample sizes got enlarged. Since the conclusion from this paper is so provoking, it is highly recommended that the authors to enlarge their cohort and perform the same analysis.

2. Sequence depth. Although a large amount of data was collected (1.5×10^8 reads), these reads were from a large amount of cells (5.8×10^7 cells). The saturation level is relatively low. At least, the saturation level need to be tested with randomized sun-sampling strategy. When dealing with diverse repertoire such as the one from naïve T cells, it is crucial to reach the saturation or near-saturation level before conclusions, such as diversity, can be made.

3. Similar to #2, it is not clear which strategy were taken to deal with systemic and experimental error. To study diverse repertoire such as the one from naïve T cells, this is also crucial.

Specific Concerns:

4. The sequencing results showed that the Tcm cells from patients were significant more diverse and less clonal. This reviewer can picture this scenery for naïve T cells, which does not have antigen experience. However, for Tcm cells formed after antigen experiences, if there is chronic autoantigen stimulation in patients, why there is no clonal selection of Tm, which should result in less diversity. The authors need to discuss this issue.

5. The authors stated that Tregs have different thymic origin to that of Tconv cells. This is not true anymore based on most recently published lineage tracking and repertoire analysis. Even for thymic Tregs, a very significant portion of them were homing from periphery. To really interpret their Treg data, the authors can track the common clones between sequenced Tn, Tm, and Treg pool.

6. With current sample size, the shortening of CDR3 length (1 a.a.) in Tn, Tcm, Treg, and Tscm from T1D patients can only be called moderate, not significant. With this kind of difference, it is difficult to imagine, structure-wise, that it "could lead to a higher degree of sequence sharing, as with shorter sequences, the chance of two TCRB CDR3s being identical increases". Again, sample size is too small to make this claim.

7. The authors stated: "In summary, we have shown that type 1 diabetes patients present alterations in the pre-selection TCRB repertoire, including an increased frequency of shorter TCRB CDR3s, which undergo enrichment during positive selection." The authors need to discuss why the impact of negative selection was excluded.

8. Page 12, "These 248 clonotypes were found in TN cells (T1D-exclusive: 76.5%. HD-exclusive: 77.8%), CM cells 249 (T1D-exclusive: 18.7%. HD-exclusive: 14.3%), Treg cells (T1D-exclusive: 2.56%. HD-exclusive: 250 3.74%) and Tscm cells (T1D-exclusive: 2.18%. HD-exclusive: 4.11%)." The authors should analyze whether these differences were HLA-related, which could strengthen their conclusion.

9. "T1D-exclusive clonotypes differed in several respects from their control counterparts. First, they showed higher frequencies, mainly in the TN and CM pools (Fig. 7a), suggesting that they have undergone more rounds of expansion." This is puzzling. Why naïve T cells went for more rounds of proliferation? If so, how can a more diverse repertoire be observed?

We thank the Editor and Reviewers for their helpful comments on our manuscript "T-cell receptor β chains show abnormal shortening, repertoire diversity and sharing in type 1 diabetes". Specifically, the editorial feedback suggested that we should perform further experiments if required to address all of the reviewers' criticisms, **particularly over whether the observed TCR sequence abnormality actually causes changes in TCR-MHC+peptide interaction or the extent of autoreactivity**. As a consequence of these and other comments, we have performed additional experiments which (i) extend the number of subjects in the sequencing study (from n=16 to n=28) to obtain a more robust dataset and (ii) which address the question of the extent to which TCR sequence abnormality impacts upon autoreactivity. To be more precise, the extra studies on TCRB shortness and autoreactivity, which involved generating and TCR sequencing a large number (532) of antigen-specific clonotypes, have focused on what we consider to be our most impactful and robust finding – the association between disease and short TCRB CDR3s. The impact of this finding relates to its novelty and potential for opening up a new area of study in relation to autoimmune disease risk. The robustness of the finding relates to the fact that all significance levels for comparisons of TCRB length were extended when we added in the new subjects.

To summarise our view, we feel that we are able to respond to all of the Reviewers' comments appropriately, including through the introduction of an extensive set of new subjects/samples (bolstering the TCRB length data) and an entirely new set of studies (providing a link between short TCRBs and autoreactivity).

Specific comments to Reviewers are below:

Comment number	Reviewer #1 (Remarks to the Author):	AUTHOR RESPONSE
R1_#1	The main concern with the current conclusions is that the methods are analysing TCR gene usage from cDNA rather than gDNA of sorted cell subsets. This practice is then susceptible to alterations in TCR gene expression per cell, with large variance possible from quiescent T cells to those of highly activated subsets that could dramatically skew the results and data interpretation. This is particularly important when comparing T cells from autoimmune donors to those of normal healthy controls. The authors should clarify and/or provide some data to clearly demonstrate that the results they have observed are in fact a general phenomenon observable	The reviewer's comment is well taken. Material was not available from our original study to enable a direct comparison of cDNA versus gDNA. Therefore to address this comment directly, we have performed a comparison between gDNA and cDNA sequencing on a set of freshly obtained samples – TN and CM cells from 3 healthy donors were sorted, RNA and gDNA isolated, cDNA synthesized, and cDNA and gDNA deep sequenced to compare similarities of the TCRB repertoires using these different nucleic acid materials from the same cells. In comparing gDNA and cDNA we are able to demonstrate a very high level of repertoire sharing, as evidenced by the overlap indices (Supplementary Figure 3). The methods are described on pages 25-26 and results discussed on page 6, 1 st paragraph. These new data indicate that the use of cDNA is highly representative of the repertoire revealed using gDNA. We agree that the use of cDNA/mRNA could be susceptible to the influence of TCR gene expression levels, although it is hard to see how this

	at the gDNA level from sorted T cell subsets where one read is equivalent to one clone.	would impact upon our major finding (higher frequency of shorter TCRBs) especially as this finding now has additional supportive evidence (see response to Reviewer 2, #1). Nonetheless, we consider it important that studies which build on our findings are aware of the potential influence of the selected technology, and we have therefore included a consideration of these points in the Discussion (Pages 20 and 21, last and 1 st paragraph, respectively).
R1_#2	-The discussion of hydrophobicity at specific residues (positions P6/P7) promoting autoreactivity is largely extrapolated from animal model data or from limited autoreactive human T cell clones. Therefore, the discussion of this topic should be somewhat more guarded than what is currently presented that suggests this finding is a universally accepted phenomenon.	This point is well taken. We have toned down the sentence that starts "It could be proposed that TCR repertoires..." (page 4, 2 nd paragraph); introduced more equivocation into the results section (page 13, 2 nd paragraph); and in the Discussion (page 20, 2 nd paragraph). We have also downplayed this aspect by moving the Figure on amino acid usage to Supplementary.
R1_#3	-While the number of subjects is small and may preclude analysis at this time, the authors should discuss what genetic risk variants, epigenetic processes, and/or developmental processes may account for this proposed universal skewing of the repertoire observed. Is this phenomenon observed in at-risk double autoantibody positive subjects prior to overt disease? Can this TCR analysis discriminate type 1 diabetes subjects from controls?	As discussed, numbers have been extended. Possible epigenetic and developmental effects are discussed (Discussion section, page 19 and 21, 1 st and last paragraphs, respectively).
R1_#4	-What are the statistical tests that were run for the conclusions that are stated in Figure 2?	Statistical tests have been added to the corresponding figure caption.
R1_#5	-The type 1 diabetes cohort appeared to have a number of older individuals. Is this phenomenon also observed in young type 1 diabetes donors that tend to exhibit a more rapid progression of the disease and loss of residual C-peptide?	We have added a comment regarding the need to perform these studies in relation to T1D progression (Discussion section, page 21, last paragraph). In the present study the size of blood volume required precluded children and adolescents being studied, but now we have an idea where to focus our questions, they can be included in future designs.
R1_#6	-Table 1, limit to appropriate number of significant figures	Table 1 has been moved to Supplementary.
R1_#7	-Correct typo in legend of Figure	This change is made as requested.

	1	
R1_#8	-Figure 2 scale bar and text is illegible in its current form. In addition, the panels could be more clearly labeled with cohort designations and AA or nucleotide information. Clarify statistical tests for e-h panels and display data from the whole cohort summarized.	These changes are made as requested.
R1_#9	-Figure 3 stats should be clarified for b-c	These changes are made as requested.
	Reviewer #2 (Remarks to the Author):	
R2_#1	 It is not clear to me if these clonotypes with shorter CDR3 length interact MHC/peptide complex and if they have anything to do with the immunopathology. In that respect, no data demonstrates that the abnormal TCRB repertoire influence the self recognition by T cells and if the CD4+T cell bearing these TCRB chain with shorter CDR3 loops interact more efficiently or less with MHC/peptide. 	This point is also the one highlighted in the Editor's covering note. Addressing this through a full structural analysis of the efficiency of TCRs that differ in CDR3B length and bind the same peptide/MHC is far beyond being an adjunct to the present analysis. We therefore elected to address this issue through a different route, albeit one that required extensive new studies. We asked the question – do CD4⁺ T cells that are known to be autoreactive have short TCRBs in comparison to anti-viral TCRBs or to the “normal” TCRB length distribution. If true, this would provide further compelling evidence that TCRB bias is an important determinant of autoreactivity. We addressed this in two different ways, first by sequencing 532 TCRBs from a variety of auto- and viral-reactive cells we generated; and second by amassing ALL of the autoreactive TCRB data available for T1D patients from the literature. Importantly, in these studies we find that autoreactive clonotypes are shorter than viral ones, and fall in the shorter spectrum of distribution of TCRB CDR3 lengths inferred by deep sequencing of healthy donor repertoires. Moreover, length analysis of all T1D autoreactive clonotypes described in the literature indicates that indeed they are shorter than normal TCRB CDR3 lengths. This length analysis even appears to distinguish T1D patients from non-diabetic autoantibody-positive subjects. In summary, we conclude that this major component of our discovery – short TCRBs in T1D – has now acquired a new and compelling dataset to suggest that it can influence self-recognition by

		CD4⁺ T cells. These findings are described in Methods (pages 29-31) and Results (pages 13-15) and we consider them of sufficient importance that they have been commented upon in the revised Abstract and Discussion.
R2_#2	 The authors try to explain the mechanism through which more V/D/J deletion might arise. If, as stated by the authors, the presence of TCRβ chain with shorter CDR3 loops is due to an alteration of VDJ recombination complex, this should be found in whole lymphocyte receptors chains. Therefore, it would also be interesting to check if the CD8+T cells TCRβ chains exhibit the same defects. 	We agree that this proposed extra study would be of interest (as mentioned in the Discussion section, page 21, last paragraph), but given our focus in the additional work that we conducted (to provide additional data on robustness of the findings and to analyse the impact of short TCRBs) we feel that studying CD8⁺ T cells is beyond the scope of the current manuscript.
R2_#3	 Somehow these results are not original, many publications are already reported that type 1 diabetes patients display a disturbed T cell repertoire and as mentioned by the authors both shortening of TCRB CDR3 length and sharing clonotypes are already been observed in patients with others autoimmune diseases. 	We do not entirely agree with the Reviewer here. We already cited the study on Rasmussen's encephalitis, which has some similarities (but actually did not include any discussion of the reason behind the disturbed CDR3B length); we are not aware of any study that has sequenced anywhere near this number of subjects to anywhere near this depth. Given that T1D is the prototypic organ-specific autoimmune disease, and given the novelty of the findings we consider them worthy of reporting. Comments clarifying the novelty of the work are now included (page 16, 1st paragraph).
	Reviewer #3 (Remarks to the Author):	
R3_#1	1. Sample size. All major conclusions were made based on a rather small sample size. The previous experience in the field is that many abnormal features identified in the TCR repertoire, such as length difference, V/J usage difference, or amino acid bias, disappeared when tested sample sizes got enlarged. Since the conclusion from this paper is so provoking, it is highly recommended that the authors to enlarge their cohort and perform the same analysis.	We have increased the sample size to almost double. Importantly, all of the main abnormal features that we originally described, including the main finding of length differences and TCR rearrangement alterations, together with sharing and amino acid bias, are retained in the extended cohorts, suggesting that what we describe is robust and generalizable.
R3_#2	2. Sequence depth. Although a large amount of data was	We agree with the reviewer as to the importance of such testing. To examine whether observed

	collected (1.5X10e8 reads), these reads were from a large amount of cells (5.8X10e7 cells). The saturation level is relatively low. At least , the saturation level need to be tested with randomized sun-sampling strategy. When dealing with diverse repertoire such as the one from naïve T cells, it is crucial to reach the saturation or near-saturation level before conclusions, such as diversity, can be made.	differences (such as those in TCRB CDR3 length) are not an artefact of sub-sampling of TN cells and variable number of reads among samples, we performed randomized sub-sampling analysis and re-analyzed the TCRB CDR3 length distributions. These studies show that, even in the subsamples, type 1 diabetes patients continue to demonstrate a statistically significant reduction in TCRB CDR3 length. A new figure illustrating this has been added (Supplementary Figure 5). We have described the above in the Results section (page 7, 2 nd paragraph) and in the Material and Methods section (pages 31 and 32, last and 1 st paragraph, respectively).
R3_#3	3. Similar to #2, it is not clear which strategy were taken to deal with systemic and experimental error. To study diverse repertoire such as the one from naïve T cells, this is also crucial.	The strategy to deal with systemic and experimental error is now further described in the Material and Methods Section (page 24, 2 nd and 3 rd paragraphs., and page 25, 1 st paragraph).
R3_#4	4. The sequencing results showed that the Tcm cells from patients were significant more diverse and less clonal. This reviewer can picture this scenery for naïve T cells, which does not have antigen experience. However, for Tcm cells formed after antigen experiences, if there is chronic autoantigen stimulation in patients, why there is no clonal selection of Tm, which should result in less diversity. The authors need to discuss this issue.	When the sample size is increased the same trends on diversity are kept- however some p-values are now higher and fail to reach conventional levels of significance, and we have therefore toned down our statements on interpretation of the diversity results that the Reviewer alludes to (see Results section and discussion). Given that diseases such as type 1 diabetes may exhibit heterogeneity, studies involving more subjects will be needed to build upon the trends we report. Importantly, even with the additional subjects we confirm that CM and TN diversity are strongly correlated, so we must assume that a diverse TN repertoire promotes a diverse CM; the mechanisms underlying this will require future study.
R3_#5	5. The authors stated that Tregs have different thymic origin to that of Tconv cells. This is not true anymore based on most recently published lineage tracking and repertoire analysis. Even for thymic Tregs, a very significant portion of them were homing from periphery. To really interpret their Treg data, the authors can track the common clones between sequenced Tn, Tm, and Treg pool.	We appreciate the comment of the reviewer and have deleted the statement. We have added a sentence in the Results section (page 7, last paragraph) and Discussion section (page 19, 3 rd paragraph) , discussing the finding of a high degree of sharing of TCRB nucleotide sequences between CM and Treg cells, suggesting that Treg cells can indeed potentially be peripherally derived.
R3_#6	6. With current sample size, the shortening of CDR3 length (1 a.a.)	We have modified this section to state “moderate” as suggested (Results section, page 7, 2 nd

	in Tn, Tcm, Treg, and Tscm from T1D patients can only be called moderate, not significant. With this kind of difference, it is difficult to imagine, structure-wise, that it “could lead to a higher degree of sequence sharing, as with shorter sequences, the chance of two TCRB CDR3s being identical increases”. Again, sample size is too small to make this claim.	paragraph), although this shortening remains a highly reproducible finding and even more so after increasing the sample size. We have also further explained the relationship between high degree of sharing and length (Results section, page 7, 2nd paragraph)
R3_#7	7. The authors stated: “In summary, we have shown that type 1 diabetes patients present alterations in the pre-selection TCRB repertoire, including an increased frequency of shorter TCRB CDR3s, which undergo enrichment during positive selection.” The authors need to discuss why the impact of negative selection was excluded.	The reviewer’s point is well taken. We were intending to refer to thymic selection as a whole. Therefore, the quoted sentence has been modified to state “which undergo enrichment during thymic selection” (Results section, page 10, 2nd paragraph).
R3_#8	8. Page 12, “These 248 clonotypes were found in TN cells (T1D-exclusive: 76.5%. HD-exclusive: 77.8%), CM cells 249 (T1D-exclusive: 18.7%. HD-exclusive: 14.3%), Treg cells (T1D-exclusive: 2.56%. HD-exclusive: 250 3.74%) and Tscm cells (T1D-exclusive: 2.18%. HD-exclusive: 4.11%).” The authors should analyze whether these differences were HLA-related, which could strengthen their conclusion.	It is unclear to us what the Reviewer refers to here. This listing of the derivation of the T1D-and HD-exclusive clonotypes (“T1D- and HD-enriched” in the revised version) is intended to show that the data are not biased simply by coming from a particular cell compartment. We have performed quite extensive analyses of HLA effects and not observed any biases.
R3_#9	9. “T1D-exclusive clonotypes differed in several respects from their control counterparts. First, they showed higher frequencies, mainly in the TN and CM pools (Fig. 7a), suggesting that they have undergone more rounds of expansion.” This is puzzling. Why naïve T cells went for more rounds of proliferation? If so, how can a more diverse repertoire be observed?	This was a slightly misleading statement, for which we apologise, and which has been corrected to state “First, they showed higher frequencies, suggesting that they are more frequent in blood, and may have undergone more rounds of expansion (in the case of CM) or were common and seeded at higher frequency (in the case of TN).” (Results section, page 12, last paragraph).

Reviewers' comments:

Reviewer #1 (Remarks to the Author):

The authors conducted a significant revision to the original submission and the resulting manuscript has been bolstered considerably in terms of cohort size. The authors provide a great resource of data and information to the field regarding the parameters of the repertoire from the carefully sorted T cell subsets.

There remains, however, some concern regarding the finding of lower CDR3 lengths in the type 1 diabetes cohort. The validation data (Supplemental Figure 3) conducted in normal healthy controls (n=3) does not address the potential for rare populations of potentially highly activated T cells in T1D subjects contributing to the observed bias. The authors attempted to again limit this potential confounder by restricting the data to unique sequences, which would presumably help to eliminate expanded activated memory T cell clonotypes. The authors should carefully consider the potential technical assay parameters that may contribute to a false bias in this assay readout.

There are a couple minor typos in the text that should be addressed by careful review and editing.

Reviewer #3 (Remarks to the Author):

Most of my concerns are addressed in the revised version, except for point #3. TCR repertoire sequencing can not tolerate any random or systemic error at the CDR3 region. During the data processing of TCR repertoire sequencing, the challenge is, always, to deal with those low abundance sequences with high similarity to others. The authors need to detail the strategy and parameter used to distinguish whether highly similar CDR3s are generated by error. This is extremely important for interpretation of naive and memory T cell diversity because their clonotypes are in this low abundance range.

Specific comments to Reviewers are below:

Reviewer #1:

The authors conducted a significant revision to the original submission and the resulting manuscript has been bolstered considerably in terms of cohort size. The authors provide a great resource of data and information to the field regarding the parameters of the repertoire from the carefully sorted T cell subsets.

There remains, however, some concern regarding the finding of lower CDR3 lengths in the type 1 diabetes cohort. The validation data (Supplemental Figure 3) conducted in normal healthy controls (n=3) does not address the potential for rare populations of potentially highly activated T cells in T1D subjects contributing to the observed bias. The authors attempted to again limit this potential confounder by restricting the data to unique sequences, which would presumably help to eliminate expanded activated memory T cell clonotypes. The authors should carefully consider the potential technical assay parameters that may contribute to a false bias in this assay readout.

There are a couple minor typos in the text that should be addressed by careful review and editing.

We thank the reviewer for this comment. We have decided to perform additional analyses to address this and to devote a section of the Discussion to addressing potential technical biases as follows:

1. We performed additional analyses to examine T cell activation levels (quantified as CD25 staining by flow cytometry) on the overall TN and CM populations that were sorted. We have also focused this analysis onto the top 2% of activated cells down to the 99.9th centile (i.e. examining whether cells present at low frequency are more activated in one sample set than the other). We further examined these subsets for levels of CD3 expression as a surrogate of TCR expression. In each of these analyses we see no differences between healthy donors and type 1 diabetes patients. Thus, as far as we can tell from these studies, in our sample set there is no evidence of a bias due to the presence of rare, highly activated clones. These data are now discussed in the Results section (Page 6, first paragraph) and included as Supplementary Figure 4.
2. The Referee comments spurred us to include an expanded section in the Discussion (Pages 22 and 23, last and first paragraphs, respectively), to consider what technical biases could influence our results and how we have addressed them. The main potential biases are: variation in TCR gene expression levels per cell combined with rare, highly activated clones (see 1. above); analytical bias; and possible PCR skewing. In each case we discuss why we do not consider the data to be biased. In particular, we would highlight the use of unique clonotypes in the key analysis of length distributions, which the referee agrees is an important step in obviating bias; the lack of evidence of differences in activation levels in rare, highly activated cells from our new analyses; and the unlikely event that PCR bias could play a role.

As a final note we have re-read the manuscript carefully for correction of typos.

Reviewer #3:

Most of my concerns are addressed in the revised version, except for point #3. TCR repertoire sequencing can not tolerate any random or systemic error at the CDR3 region. During the data processing of TCR repertoire sequencing, the challenge is, always, to deal with those low abundance sequences with high similarity to others. The authors need to detail the strategy and parameter used to distinguish whether highly similar CDR3s are generated by error. This is extremely important for interpretation of naïve and memory T cell diversity because their clonotypes are in this low abundance range.

We apologize for not providing a full description of this step in data processing in previous versions (we only provided citations in support of the methods). In fact, there are strategies in place at both the sample sequencing and the data processing stages to address the issue of low abundance sequences with high similarity to others. These strategies are now described in greater detail (Methods section, Pages 25 and 26, last and first paragraphs, respectively), with cited references 26, 83 and 84.

REVIEWERS' COMMENTS:

Reviewer #1 (Remarks to the Author):

The authors have sufficiently addressed the comments. It will be important for the field to replicate these findings with other platforms and technologies, and in addition, investigate the molecular mechanisms underlying these broad alterations.

Reviewer #3 (Remarks to the Author):

My concerns were addressed in the modified version. I recommend this manuscript to be published.

Response to reviewers

Comments:

Reviewer #1 (Remarks to the Author):

The authors have sufficiently addressed the comments. It will be important for the field to replicate these findings with other platforms and technologies, and in addition, investigate the molecular mechanisms underlying these broad alterations.

Reviewer #3 (Remarks to the Author):

My concerns were addressed in the modified version. I recommend this manuscript to be published.

Response:

We would like to thank the reviewers for their comments, stating that we have addressed all issues raised by them. Therefore, no further responses are required.